Simplifying the Centrolene buckleyi complex (Amphibia: Anura: Centrolenidae): a taxonomic review and description of two new species

http://orcid.org/0000-0003-3655-2678 Franco-Mena Daniela 1 daniellafrancomena@gmail.com
De la Riva Ignacio 2
http://orcid.org/0009-0003-7110-2872 Vega-Yánez Mateo A. 1 3
http://orcid.org/0000-0002-9252-1596 Székely Paul 4 5 6
http://orcid.org/0000-0003-2638-4068 Amador Luis 7 8
Batallas Diego 1 9
Reyes-Puig Juan P. 3 10
http://orcid.org/0000-0002-6132-2738 Cisneros-Heredia Diego F. 3 11
http://orcid.org/0000-0001-9303-4346 Venegas-Valencia Khristian 12
Galeano Sandra P. 12
http://orcid.org/0000-0003-2211-6605 Culebras Jaime 13 14
http://orcid.org/0000-0003-0098-978X Guayasamin Juan M. 1 jmguayasamin@usfq.edu.ec
1 Laboratorio de Biología Evolutiva, Instituto BIOSFERA, Colegio de Ciencias Biológicas y Ambientales COCIBA, U niversidad San Francisco de Quito , Quito , Ecuador
2 Department of Biodiversity and Evolutionary Biology, Museo Nacional de Ciencias Naturales-CSIC , Madrid , Spain
3 Unidad de Investigación, Instituto Nacional de Biodiversidad (INABIO) , Quito , Ecuador
4 Museo de Zoología, Universidad Técnica Particular de Loja , Loja , Ecuador
5 Laboratorio de Ecología Tropical y Servicios Ecosistémicos (EcoSs-Lab), Facultad de Ciencias Exactas y Naturales, Departamento de Ciencias Biológicas y Agropecuarias, Universidad Técnica Particular de Loja , Loja , Ecuador
6 Research Center of the Department of Natural Sciences, Faculty of Natural and Agricultural Sciences, Ovidius University Constanţa , Constanța , Romania
7 Museum of Southwestern Biology and Department of Biology, University of New Mexico , Albuquerque, NM , USA
8 Instituto BIOSFERA, Universidad San Francisco de Quito , Quito, Pichincha , Ecuador
9 Departamento de Biodiversidad, Ecología y Evolución de la Facultad de Ciencias Biológicas, Programa de Doctorado en Biología, Universidad Complutense de Madrid , Madrid , Spain
10 Fundación Oscar Efrén Reyes, Departamento de Ambiente, Fundación EcoMinga , Baños , Ecuador
11 Laboratorio de Zoología Terrestre & Museo de Zoología, Instituto de Biodiversidad Tropical IBIOTROP, Colegio de Ciencias Biológicas y Ambientales, Universidad San Francisco de Quito USFQ , Quito , Ecuador
12 Centro de Colecciones y Gestión de Especies, Instituto de Investigación de Recursos Biológicos Alexander von Humboldt , Villa de Leyva, Boyacá , Colombia
13 Photo Wildlife Tours , Quito , Ecuador
14 Fundación Cóndor Andino , Quito , Ecuador
Provete Diogo
Electronic publication date: 2024 Aug 20
Publication date: 2024
Volume: 12
Electronic Location ID: e17712
Received 2023 Sep 28; Accepted 2024 Jun 18
Copyright: © 2024 Franco-Mena et al.
Copyright year: 2024
Copyright holder: Franco-Mena et al.
License: This is an open access article distributed under the terms of the Creative Commons Attribution License, which permits unrestricted use, distribution, reproduction and adaptation in any medium and for any purpose provided that it is properly attributed. For attribution, the original author(s), title, publication source (PeerJ) and either DOI or URL of the article must be cited.
License URL: https://creativecommons.org/licenses/by/4.0/

Keywords: Andes, Anura, Biogeography, Glassfrogs, New species, Systematics

Funding: Ecuadorian Science Agency SENESCYT INEDITA PIC-20-INE-USFQ-001 USFQ HUBI 5466, 17857, 16871 PGC2018-097421-B-I00 MCIN/AEI/10.13039/501100011033 European Union Critical Ecosystem Partnership Fund CEPF-108984 Rainforest Trust and Naturaleza y Cultura Internacional PROY_INV_BA_2022_3502 PY3502 Universidad Técnica Particular de Loja PROY_INV_BA_2022_3573 Centro Jambatu BBC Studios NHFP Fundación Cóndor Andino This work was supported by Inédita Program from the Ecuadorian Science Agency SENESCYT (Respuestas a la Crisis de Biodiversidad: La Descripción de Especies como Herramienta de Conservación; INEDITA PIC-20-INE-USFQ-001), USFQ (Grants HUBI 5466, 17857, 16871). Ignacio De la Riva was funded by project PGC2018-097421-B-I00 (PI: I. De la Riva), funded by MCIN/AEI/10.13039/501100011033 and by “ERDF A way of making Europe”, by the “European Union”. The work of Paul Székely in Abra de Zamora was funded by Critical Ecosystem Partnership Fund through the “Amphibian Conservation in the Abra de Zamora Key Biodiversity Area of Ecuador” project (CEPF-108984), Rainforest Trust and Naturaleza y Cultura Internacional through the “Investigación científica, para la protección y monitoreo de especies de anfibios en el Área Clave de Biodiversidad Abra de Zamora” project (PROY_INV_BA_2022_3502 PY3502) and by Universidad Técnica Particular de Loja through the “Fortalecimiento a la gestión del Abra de Zamora, un área clave para la conservación de la biodiversidad y los servicios ambientales de los Andes sur de Ecuador” project (PROY_INV_BA_2022_3573). The work of Jaime Culebras was funded by Centro Jambatu, BBC Studios NHFP and Fundación Cóndor Andino. There was no additional external funding received for this study. The funders had no role in study design, data collection and analysis, decision to publish, or preparation of the manuscript.

==============================
Centrolenidae is a Neotropical family widely distributed in Central and South America, with its species richness concentrated in the tropical Andes. Several taxonomic problems have been identified within this family, mostly related to species with broad geographic distributions. In this study, we assessed and redefined the species boundaries of the Centrolene buckleyi species complex, and formally described two new species from the Andes of Ecuador. These new taxa are recognized by a combination of morphometric, osteological, acoustic, and genetic data. Following IUCN criteria, we propose that the two new species should to be considered as Endangered (EN), mainly because of their small distributions and habitat loss. The C. buckleyi complex provides insights into the biogeography of closely related Andean species. As in other glassfrogs, speciation in Centrolene seems to be mediated by the linearity of the Andes, where gene flow can be restricted by topography and, also, local extinctions.

Introduction

Centrolenidae (Taylor, 1951) contains 163 recognized species (Frost, 2023). The monophyly of the family is supported by molecular characters, morphological, and behavioral traits (Guayasamin et al., 2009, 2020). Centrolenidae is distributed throughout the Neotropics, including Central America, the Andes, Amazonia, and Brazilian Atlantic Forest, with a peak in species richness concentrating in the tropical Andes (Guayasamin et al., 2020).

Most Andean glassfrogs have relatively small distributions (Guayasamin et al., 2020); a conspicuous exception in this regard is Buckley’s glassfrog, Centrolene buckleyi (Boulenger, 1882), a species with a large distribution in the tropical Andean ecoregion (Frost, 2023). Species with large distributions often represent species complexes, especially in topographically and ecologically complex areas such as the Andes (e.g., Sanín et al., 2022). Thus, it is not surprising that recent studies have documented morphological, acoustic, and molecular differences within what was traditionally recognized as C. buckleyi species complex (see Guayasamin et al., 2006a, 2008; Amador et al., 2018; Guayasamin et al., 2020). Perhaps the most surprising result is that the C. buckleyi complex is not monophyletic (Guayasamin et al., 2006b, 2020; Amador et al., 2018) and that vocalizations are strikingly different among some of the populations (Bolívar, Grant & Osorio, 1999; Guayasamin et al., 2006b, 2020). Although these differences within the C. buckleyi complex have been known for some time now, no comprehensive taxonomic analysis has been performed so far. Herein, we present an integrative revision of the C. buckleyi complex through a broad population and geographic sampling, redefine C. buckleyi, and formally describe two new species to science, until now hidden within the complex. We also discuss the biogeographic patterns of the genus Centrolene.

Materials and Methods

Ethics statement

This research was conducted under permits MAE-DNB-CM-2018-0105, MAE-DNB-CM-2015-0016, and MAATE-cmarg-2022-0575, issued by the Ministerio del Ambiente, Agua y Transición Ecológica (MAATE), Ecuador. The study was carried out in accordance with the American Society of Ichthyologists and Herpetologists following the guidelines for the use of live amphibians and reptiles in field research (Beaupre et al., 2004). Artificial intelligence was not used to generate any text in this study.

Taxonomy and species concept

We follow the taxonomy proposed by Guayasamin et al. (2009). For recognizing species, we adhere to the conceptual framework developed by Simpson (1961), Wiley (1978), and de Queiroz (2005, 2007). Conceptually, we agree with de Queiroz unified concept that defines species as a metapopulation lineage evolving separately from others (de Queiroz, 2007). Also, we can infer species boundaries based on an integrative delimitation, including molecular, vocalizations and morphological lines of evidence (de Queiroz, 2005, 2007; Padial et al., 2010).

Morphological data

Morphological characterization follows Cisneros-Heredia & Mcdiarmid (2007) and Guayasamin et al. (2009). Webbing nomenclature follows Savage & Heyer (1967), as modified by Guayasamin et al. (2006a). We examined alcohol-preserved specimens from the collection at Centro Jambatu (CJ); Herpetología, Museo de Historia Natural Gustavo Orcés V., Escuela Politécnica Nacional (MEPN-H); Instituto Nacional de Biodiversidad (INABIO); Museo de Zoología, Universidad Tecnológica Indoamérica (MZUTI); Museo de Zoología, Universidad Técnica Particular de Loja (MUTPL); Museo de Zoología, Universidad San Francisco de Quito (ZSFQ); in Ecuador, and Colección de Anfibios, Instituto de Investigación de Recursos Biológicos Alexander von Humboldt (IAvH-Am), in Colombia; all examined specimens are listed in Appendix S1. Morphological measurements were taken with a Mitutoyo® digital caliper to the nearest 0.1 mm, as described by Guayasamin & Bonaccorso (2004) and Guayasamin et al. (2022), and are as follows: (1) snout–vent length (SVL) = distance from tip of snout to posterior margin of vent; (2) femur length (FEL) = distance from cloaca to knee; (3) tibia length (TL) = length of flexed leg from knee to heel; (4) foot length (FL) = distance from proximal margin of outer metatarsal tubercle to tip of toe IV; (5) head length (HL) = distance from tip of snout to posterior angle of jaw articulation; (6) head width (HW) = width of head measured at level of jaw articulation; (7) interorbital distance (IOD) = shortest distance between upper eyelids, a measurement that equals to the subjacent frontoparietal bones; (8) eye diameter (ED) = distance between anterior and posterior borders of the eye; (9) tympanum diameter (TD) = distance between anterior and posterior borders of tympanic annulus; (10) arm length (AL) = length of flexed forearm from elbow to proximal edge of Finger I at the level of articulation with arm; (11) hand length (HAL) = distance from proximal edge of Finger I to tip of Finger III; (12) Finger I (FI) = distance from outer margin of hand to tip of Finger I; (13) Finger II (FII) = distance from outer margin of hand to tip of Finger II; (14) width of Finger III (FIII) = maximum width of Finger III measured at distal end; (15) width of Toe III (TIII) = maximum width of Toe III measured at distal end; (16) internarial distance (IND) = distance between inner edges of the nostrils; and (17) eye–nostril distance (END) = distance between the anterior edge of the eye and posterior edge of the nostril. With the measurements obtained from male data (female data were scarce) of Centrolene buckleyi species complex, and to understand morphological variations among species, we implemented a data transformation to reduce the effect of allometry (Lleonart, Salat & Torres, 2000). Using the RRPP Version 1.4.0 package in R (Collyer & Adams, 2024) we performed linear regression analysis between snout-vent length (SVL) and the other 16 morphological measurements. From the residuals obtained (Table S1), we performed principal component analysis (PCA) and discriminant analysis of principal components (DAPC), using the ADEGENET package version 1.7 in R (Jombart, Devillard & Balloux, 2010).

Phylogenetic relationships

We generated 44 mitochondrial sequences of markers 12S and 16S, from 25 individuals belonging to six species of Centrolene. All new sequences were deposited in GenBank (Table 1). For DNA extractions we follow Peñafiel et al. (2019) while amplification and sequencing protocols follow Guayasamin et al. (2008). The newly obtained sequences were compared to species in the Centrolene buckleyi complex (see Amador et al., 2018) and all other glassfrog genera, downloaded from GenBank (https://www.ncbi.nlm.nih.gov/genbank/) (Appendix S1). Most previous sequences were generated by Guayasamin et al. (2008, 2020) and Amador et al. (2018). Sequences were aligned using MAFFT v.7 (Multiple Alignment Program for Amino Acid or Nucleotide Sequences: http://mafft.cbrc.jp/alignment/software/), with the G-INS-i strategy (Katoh & Standley, 2013). We used Mesquite 1.12 to visualize the alignment (Maddison & Maddison, 2019). Uncorrected pairwise genetic distances among Centrolene species were calculated with MEGA 11 (Tamura, Stecher & Kumar, 2021).

Table 1 Species, vouchers, and GenBank accession numbers for newly generated DNA sequences (12S–16S) used in genetic analyses.

Species	Museum number	12S	16S	
Centrolene buckleyi	DHMECN 867	‒	OR479083	
Centrolene buckleyi	DHMECN 13828	OR479108	OR479085	
Centrolene buckleyi	DHMECN 14180	‒	OR479086	
Centrolene buckleyi	ZSFQ 4420	OR479107	OR479084	
Centrolene buckleyi	ZSFQ 4421	OR479109	OR479087	
Centrolene buckleyi	ZSFQ 5366	OR479112	OR479090	
Centrolene buckleyi	CJ 1055	OR479115	OR479093	
Centrolene buckleyi	CJ 2171	OR479110	OR479088	
Centrolene buckleyi	CJ 9789	OR479114	OR479092	
Centrolene buckleyi	CJ 4292	OR479113	OR479091	
Centrolene buckleyi	CJ 11305	OR479111	OR479089	
Centrolene elisae sp. nov.	ZSFQ 4228	OR479117	OR479099	
Centrolene cf. elisae	ZSFQ 2134	OR479116	OR479098	
Centrolene marcoreyesi sp. nov.	CJ 11364	OR479121	OR479097	
Centrolene marcoreyesi sp. nov.	CJ 11564	OR479118	OR479094	
Centrolene marcoreyesi sp. nov.	CJ 12631	OR479120	OR479096	
Centrolene marcoreyesi sp. nov.	MUTPL 271	OR479119	OR479095	
Centrolene cf. venezuelense	IAvH-Am-17401	OR479122	OR479100	
Centrolene cf. venezuelense	IAvH-Am-17403	OR479124	OR479102	
Centrolene cf. venezuelense	IAvH-Am-17407	OR479125	OR479103	
Centrolene cf. venezuelense	IAvH-Am-17410	OR479123	OR479101	
Centrolene sp.	ZSFQ 621	OR479128	OR479106	
Centrolene sp.	ZSFQ 4422	OR479126	OR479104	
Centrolene sp.	ZSFQ 4423	OR479127	OR479105	
Note:

Acronyms are CJ, Centro Jambatu; DHMECN, División de Herpetología, Museo Ecuatoriano de Ciencias Naturales; MUTPL, Museo de Zoología, Universidad Técnica Particular de Loja; ZSFQ, Museo de Zoología Universidad San Francisco de Quito; IAvH-Am, Colección de Anfibios, Instituto de Investigación de Recursos Biológicos Alexander von Humboldt.

Phylogenetic trees were performed using maximum likelihood (ML) and Bayesian inference (BI) methods. Maximum likelihood tree was run in the software IQ-TREE 1.6.8 (Nguyen et al., 2015), node support was assessed via 10,000 ultra-fast bootstrap replicates (Minh, Nguyen & von Haeseler, 2013). Ultra-fast bootstrapping also leads to a straightforward interpretation of the support values (e.g., support of ≥95% should be interpreted as significant; Minh, Nguyen & von Haeseler, 2013). Bayesian inferences were performed in MrBayes 3.2.7 (Ronquist et al., 2012). We conducted four parallel runs of Markov Chain Monte Carlo (MCMC) for 10,000,000 generations, with sampling every 1,000 iterations and burning of 25%, to estimate the Bayesian tree and Bayesian Posterior Probabilities (BPP). Finally, all trees generated were visualized in iTol v5 (Letunic & Bork, 2021) and edited in Adobe Illustrator 15.0.0 (Adobe Systems Inc., San Jose, CA, USA). Clade support was considered significant for values ≥95 in bootstrap analyses under ML (Minh, Nguyen & von Haeseler, 2013) and with posterior probabilities ≥0.95 in BI (Huelsenbeck et al., 2001).

Osteology

Osteological images of one specimen of Centrolene buckleyi sensu stricto (MZUTI 763) and the holotypes (MZUTI 84, ZSFQ 4418) of the two new species were obtained using emission of X-rays and then were transformed to recreate a three-dimensional volumetric map of the object (Du Plessis et al., 2017) in a XTH 160 Nikon Metrology with a molybdenum target, and then raw X-ray data were elaborated using CTPro 3D software (Nikon Metrology, Brighton, MI, USA), at Museo Nacional de Ciencias Naturales-CSIC (Madrid, Spain). Morphological comparisons and visualization of the micro-CT images were performed with myVGL 3.0.4 (Volume Graphics, Heidelberg, Germany) and we added color to the micro-CT scan images using Adobe Photoshop. The osteology descriptions follow the terminology of Trueb (1973), Duellman & Trueb (1986), and Guayasamin & Trueb (2007). Cartilage structures were excluded from the osteological descriptions because the settings selected for the micro-CT images do not recover cartilaginous structures. All CT-Scans are available at the Museo Nacional de Ciencias Naturales-CSIC (Madrid, Spain).

Bioacoustics

The recordings were obtained with an Olympus LS-11 digital recorder, at a sampling frequency of 44 kHz and 16-bits resolution and saved in uncompressed WAV and aiff format. Call recordings are stored at the Laboratorio de Biología Evolutiva at Universidad San Francisco de Quito (LBE-USFQ) and Fonoteca UTPL (FUTPL). The calls were analyzed in Raven 1.6 (Lisa Yang & Center for Conservation Bioacoustics at the Cornell Lab of Ornithology, 2023), having as settings a Hann window at 50% overlap and 512 samples of FFT size. The figures were processed in R (R Core Team, 2018), using Seewave 2.2.0 (Sueur, Aubin & Simonis, 2008) and tune R 1.4.1 (Ligges et al., 2018).

Definitions, terminology, and measurements of the acoustic parameters follow Köehler et al. (2017) and Sueur (2018), with a note-centered approach (sensu Köehler et al., 2017). Call type and structure were classified according to Duarte-Marín et al. (2022) and Emmrich et al. (2020). The following temporal and spectral parameters were measured and analyzed. Call duration: time from the beginning to the end of a call; in the case of single-note calls this is the same as a note duration; inter-call interval: the interval between two consecutive calls, measured from the end of one call to the beginning of the consecutive call; call rate: number of calls/minute, measured as the time between the beginning of the first call and the beginning of the last call; notes/call: number of notes present in a call; note duration: the duration of a single note within a call, measured from beginning to the end of the note (in the case of double-note calls); inter-note interval: the interval between two consecutive notes within the same call, measured from the end of one note to the beginning of the consecutive note; note rate: number of notes per second, measured as the time between the beginning of the first note and the beginning of the last note; pulse/note: number of pulses present in a note; pulse duration: time measured from one amplitude minimum to the next amplitude minimum of a pulse; pulse rate: number of pulses/second; dominant frequency: the frequency containing the highest sound energy, measured along the entire call; the 90% bandwidth, reported as frequency 5% and frequency 95%, or the minimum and maximum frequencies, excluding the 5% below and above the total energy in the selected call; frequency modulation: change in the instantaneous frequency of a signal over time, measured as the initial dominant frequency vs. final dominant frequency of the note; number of visible harmonics and frequency of each visible harmonic. Measures of central tendency (means) and dispersion (maximum, minimum, and standard deviation) were calculated for all acoustic parameter values analyzed. Abbreviations used in the units of measurement correspond to: kilohertz (kHz); milliseconds (ms); seconds (s); per minute(/min); per second (/s).

Biogeographic history

To reconstruct the ancestral distribution of each node of the 12S–16S calibrated phylogenetic hypothesis (see analyses below) of our Centrolene data set, we used the R package BioGeoBEARS (BioGeography with Bayesian (and likelihood) Evolutionary Analysis in R Scripts) (Matzke, 2013). Specifically, we ran our biogeographic analysis considering three different models (DEC, DIVALIKE, and BAYAREALIKE) to obtain a probability distribution of the most probable ancestral areas and the diversification of species. For this analysis, we used the species delimitation scenario proposed by Amador et al. (2018) and recorded the exact geographical distribution of Centrolene species included in this analysis. We coded species distribution according to the Andean Mountain Range sections, northern Andes (Venezuela, Colombia, and Ecuador), and central Andes (Peru), using the species delimitation scenario proposed by Amador et al. (2018). We used the Akaike’s information criterion corrected (AICc) to select the best fit model. For the analysis, we used a maximum clade credibility (MCC) tree obtained with BEAST v.2.6.6 (Bouckaert et al., 2019) using secondary calibration based on the temporal calibration scheme outlined by Castroviejo-Fisher et al. (2014) who reported the origin of the most recent common ancestor (MRCA) of Centrolene 13.05 Ma (10.25–16.34). We used a relaxed clock log normal prior linked to both partitions and a calibrated Yule model of speciation as a tree prior. The analysis was run for 7 × 107 generations and were sampled every 5,000 generations. The trace log file was checked for convergence and for ESS values above 200 using Tracer v.1.7.2 (Rambaut et al., 2018). The MCC tree was estimated with Treeanotator (program distributed as part of BEAST) with the sampled trees after discarding the first 20% as burn-in. We used FigTree 1.4.4 (Rambaut, 2014) to visualize the summarized MCC tree.

Nomenclatural acts

The electronic version of this article in Portable Document Format (PDF) will represent a published work according to the International Commission on Zoological Nomenclature (ICZN), and hence the new names contained in the electronic version are effectively published under that Code from the electronic edition alone. This published work and the nomenclatural acts it contains have been registered in ZooBank, the online registration system for the ICZN. The ZooBank LSIDs (Life Science Identifiers) can be resolved, and the associated information viewed through any standard web browser by appending the LSID to the prefix http://zoobank.org/. The LSID for this publication is: urn:lsid:zoobank.org:pub:EB178068-646B-4071-96B4-C0614D90A366. The online version of this work is archived and available from the following digital repositories: PeerJ, PubMed Central SCIE and CLOCKSS.

Results

The Centrolene buckleyi species complex from a phylogenetic perspective. Both methods of phylogenetic reconstruction (ML and BI) inferred similar evolutionary relationships, regarding the lineages that, based on overall morphological similarity, are part of the C. buckleyi species complex (Fig. 1). The optimal nucleotide substitution model for our dataset according to Model-Finder (lnL = −11,309.1953; BIC = 24,016.8084) was TIM2+F+I+G4. In general, the BI tree showed higher nodal support and a lower number of collapsed nodes than the ML tree. Since both analyses resulted in identical topologies, we present the ML tree, including support values for each node obtained from both ultrafast bootstraps of ML and Bayesian posterior probability (i.e., UFB/BPP) (Fig. 1).

Figure 1 Phylogenetic relationships of species in the genus Centrolene, inferred under Maximum Likelihood criterion and based on a concatenated dataset of mitochondrial genes (12S + 16S).

Node support is expressed in Bootstrap values (%), followed by Bayesian posterior probabilities; missing values indicate support below 60 (bootstrap) or 0.6 (posterior probability). Each terminal includes the following information: species name, voucher number, and locality. New sequences generated in this study are in blue. Photographs of C. buckleyi sensu stricto by Juan M. Guayasamin, C. elisae sp. nov. by Daniela Franco-Mena and C. marcoreyesi. sp. nov. by Paul Székely.

The inferred phylogeny confirms the placement of all sampled populations in the genus Centrolene (Jiménez de la Espada, 1872), as defined by Guayasamin et al. (2009). Our inferred relationships among Centrolene species are similar to those reported in previous studies (Guayasamin et al., 2008, 2020; Twomey, Delia & Castroviejo-Fisher, 2014; Amador et al., 2018; Székely et al., 2023b; Cisneros-Heredia et al., 2023), but some novel relationships are revealed because of our increased taxon sampling (Fig. 1). Our phylogenetic analysis shows that Centrolene buckleyi represents a species complex as suggested in previous studies (Guayasamin et al.,2006a, 2020; Amador et al., 2018), with at least four undescribed species, two of which we formally describe below. We also update the description of C. buckleyi sensu stricto because the redescription of the species (Guayasamin et al., 2020) included some individuals that correspond to one of the new species described below.

Systematics.

Centrolene buckleyi (Boulenger, 1882) sensu stricto

Hylella buckleyi Boulenger, 1882

Centrolenella buckleyi—Noble, 1920

Hyla purpurea—Nieden, 1923

Cochranella buckleyi—Taylor, 1951

Centrolenella buckleyi—Goin, 1964

Centrolenella buckleyi buckleyi—Rivero, 1968

Centrolenella johnelsi—Cochran & Goin, 1970

Centrolene buckleyi—Ruiz-Carranza & Lynch, 1991

Centrolenella buckleyi—Ayarzagüena, 1992

English common name. Buckley’s Glassfrog

Spanish common name. Rana de Cristal de Buckley

Holotype. MLS 432. Type locality: “San Pedro, N of Medellín, Antioquia, Colombia” (Goin, 1961). Now destroyed (Frost, 2023).

Syntypes. BMNH 80.12.5.201; from Pallatanga, Provincia de Chimborazo, Ecuador, is now almost completely macerated in ethanol and almost no bones remain, and the other, BMNH 78.1.25.16; from Intac, Imbabura province, Ecuador, is missing (Duellman, 1977; Guayasamin et al., 2020).

Neotype. KU 202770, adult female. Collected from Laguna de Cuicocha, Imbabura province, Ecuador. See the description in Guayasamin et al. (2020).

Amended definition. (1) SVL in adult males 26.1–32.5 mm (n = 17), in females 24.2–39.8 (n = 13); (2) in life, dorsum light to dark green with our without scattered darker green patches; upper lip white, usually with a white line extending backwards along the flanks of body; bones green; (3) iris gray-white with thin black reticulation and a horizontal brown stripe; (4) humeral spines, vocal slits and sacs present in adult males; (5) snout round in dorsal aspect, sloping in lateral profile; (6) webbing absent between Fingers I–III; reduced between outer fingers: III (21/4–3−)—(2+–21/2) IV; (7) webbing formula on foot: I (11/2–2−)—(2–21/4) II (1−–1+)—(2+–21/2) III (1+–12/3)—(21/3–3) IV (22/3–3)—(12/3–2−) V; (8) ulnar fold low and white, ventrolateral margin of arm white; inner tarsal fold evident; outer tarsal fold absent, external ventrolateral margin of tarsus white; (9) prepollex concealed; in males, nuptial pad Type I; (10) Toe II slightly longer than Toe I.

Comparison with similar species. In life, Centrolene buckleyi sensu stricto is differentiated from its congeners by having dorsal surfaces light green to dark green (some individuals present scattered olive-green patches), white upper lip, inclined snout in profile, large humeral spine (in adult males), and reduced webbing between fingers (Figs. 2 and 3). Centrolene buckleyi sensu stricto (SVL = 25.0–34.7 mm) is larger than C. elisae sp. nov. (SVL = 22.0–25.3 mm) and C. marcoreyesi sp. nov. (SVL = 24.5–27.0 mm). Differences between C. buckleyi sensu stricto and morphologically similar taxa are summarized in Tables 2, 3 and Fig. 4. Main skull characters to discriminate species are summarized in Table 4. Relevant genetic distances are shown in Table S2 and Fig. S1. The advertisement calls of C. buckleyi and the two newly described species exhibit non-overlapping differences in some key traits (i.e., call duration, number of notes, and dominant frequency; see Table 5).

Figure 2 Color patterns in life of closely related glassfrogs.

(A) Centrolene buckleyi sensu stricto (adult males, MZUTI 763, DHMECN 13828, ZSFQ 4420). (B) Centrolene elisae sp. nov., paratypes (adult female, ZSFQ 5367; adult male, ZSFQ 5369; adult male, ZSFQ 4428). (C) Centrolene marcoreyesi sp. nov. (paratype, MUTPL 271, adult male). Photographs of C. buckleyi by Juan M. Guayasamin, and Diego Batallas-Revelo; C. elisae sp. nov. by Mateo Vega-Yánez and Daniela Franco-Mena; and C. marcoreyesi sp. nov. by Paul Székely.

Figure 3 Comparison of species previously confused with Centrolene buckleyi, in ethanol.

From left to right: Dorsal view, ventral view, head in dorsal view, head in lateral view, hand in ventral view, and foot in ventral view. (A) Centrolene buckleyi sensu stricto, male, MZUTI 0763; (B) C. elisae sp. nov., male holotype MZUTI-084; (C) C. marcoreyesi sp. nov. male holotype, ZSFQ 4418. Photographs by Daniela Franco-Mena (A, C), and Mateo Vega-Yánez (B).

Table 2 Differences between the new species and morphologically similar taxa within Centrolene.

Species	Localities	SVL in adult males (mm)	Snout (lateral view)	Texture of dorsal skin (males)	Dorsal coloration (in life)	Source	
C. altitudinalis	Venezuela: Andes, Estado Mérida, 1,975–2,400 m.	21.5–24.5 (n = 12)	Round to slightly sloping	Shagreen with small spicules	Uniform dark green dorsum with white or cream dots	Señaris & Ayarzagüena (2005)	
C. ballux	Ecuador, Colombia: Pacific slopes of Andes, 1,780–2,340 m.	19.2–23.3 (n = 25)	Bluntly rounded	Shagreen	Green dorsum with small
light spots	Duellman & Burrowes (1989), Guayasamin et al. (2020)	
C. buckleyi sensu stricto	Ecuador: central and northern Andes. Cordillera Oriental and Cordillera Occidental; 2,573–3,416 m.	25.0–34.7 (n = 20)	Slightly sloping to sloping	Shagreen with or without small warts	Bright to dark green; some individuals have scattered olive-green spots on the dorsum	Guayasamin et al. (2020); (J Culebras, 2022, in preparation); this study	
C. camposi	Ecuador: Southwestern slopes of the Cordillera Occidental of the Andes; 2,950 m.	29.1–31.2 (n = 2)	Sloping	Shagreen with dispersed low and rounded warts	Uniform green dorsum with light green warts	Cisneros-Heredia et al. (2023)	
C. condor	Ecuador: Eastern Montane Forest ecoregion; 1,737–2,920 m.	23.7–28.6 (n = 7)	Subacuminated	Shagreen with low warts and abundant spicules	Green with many small yellowish–white flecks and dark bluish-black/brown flecks and punctuations	Almendáriz & Batallas (2012), Cisneros-Heredia & Morales-Mite (2008)	
C. elisae sp. nov.	Ecuador: eastern versant of central and northern Andes; 2,118–2,586 m.	22.0–25.3 (n = 5)	Rounded	Shagreen covered with minute spinules and spots	Dark green with small to minute white spots.	This study	
C. ericsmithi	Ecuador: Southwestern slopes of the Cordillera Occidental of the Andes; 2,950 m.	27.3
(n = 1)	Rounded	Shagreen with dispersed spicules, and covered by microspicules	Bright green dorsum, white tubercles	Cisneros-Heredia et al. (2023)	
C. heloderma	Ecuador: Montane forests; 1,850–2,575 m	26.8–31.5 (n = 17)	Sloping	Pustular	Green with green to
bluish white warts	Duellman (1981), Guayasamin et al. (2020)	
C. hesperia	Perú: Pacific slope of the Cordillera Central; 1,500–1,800 m.	23.0–27.3 (n = 54)	Slightly sloping	Shagreen with spinules	Dorsal life leaf green with green spicules	Cadle & McDiarmid (1990)	
C. huilense	Ecuador: Amazonian
slope of the Andes; 2,100–2,190 m.	23.6–26.7 (n = 7)	Sloping	Shagreen with spinules	Green with dark green to dark lavender spots and
smaller white spots	Ruiz-Carranza, Ardila-Robayo & Lynch (1996), Guayasamin et al. (2020)	
C. lynchi	Ecuador: Pacific slope of the Cordillera Occidental of the Andes; 1,140–1,852 m.	23.3–26.5 mm (n = 22)	Round	Shagreen with spinules	Dull green with minute yellowish–white warts.	Duellman & Burrowes (1989), Guayasamin et al. (2020)	
C. marcoreyesi sp. nov.	Ecuador: Eastern slopes of the southern Andes; 2,008–2,923 m.	24.5–27.0 (n = 6)	Sloping	Shagreen with dispersed low warts	Green, with whitish spots	This study	
C. muelleri	Peru: Huallaga and Marañón drainages in the southern; 1,830–2,050 m.	23.5	Slightly sloping	Finely shagreen with dorsolateral rows of warts	Green with dark greenish-black spots and pale-yellow tubercles	Duellman & Schulte (1993), Guayasamin et al. (2020)	
C. notosticta	Colombia: Western slope of the Eastern Andes; 1,661–2,440 m.	19.4–22.7 (n = 31)	Blunt	Shagreen with spinules	Green with small yellow spots	Ruiz-Carranza & Lynch (1991)	
C. sabini	Peru: Kosñipata valley; 2,750–2,800 m.	29.6–31.2 (n = 5)	Obtuse	Skin on dorsal surfaces of head and body spiculate; skin on dorsal surfaces of limbs smooth.	Green with yellowish-green spots and
patches	Catenazzi et al. (2012)	
C. venezuelense	Venezuela: Andean mountains; 2,400–3,050 m.	23.4–33.8 (n = 15)	Rounded	Dorsum with smooth to finely granular skin, with spicules of different sizes.	Light green, with small cream-colored spots.	Señaris & Ayarzagüena (2005)	
C. zarza	Ecuador: Southern montane forest; 1,434–1,480 m.	23.2–26.2 (n = 5)	Sloping	Shagreen with elevated, and some enameled, warts	Light green with many white or whitish spots and flecks	Székely et al. (2023b)	

Table 3 Morphological measurement of Centrolene buckleyi sensu stricto, C. elisae sp. nov., and C. marcoreyesi sp. nov.

	C. buckleyi	C. elisae sp. nov.	C. marcoreyesi sp. nov.	
	Males (n = 17)	Females
(n = 13)	Males
(n = 6)	Females (n = 1)	Males
(n = 6)	
SVL	26.1–32.5 (27.9 ± 1.5)	24.2–39.8 (28.7 ± 4.3)	22.9–25.3 (24.3 ± 0.8)	27.2	24.5–27.0 (25.9 ± 1.0)	
FEL	14.0–16.9 (15.1 ± 1.0)	13.3–19.6 (15.2 ± 1.9)	13.1–14.7 (13.5 ± 0.6)	15.3	13.7–15.1 (14.3 ± 0.6)	
TL	14.3–18.3 (15.6 ± 1.2)	13.7 –18.6 (15.5 ± 1.5)	13.9–15.1 (14.6 ± 0.4)	16.4	15.3–15.8 (15.6 ± 0.2)	
FL	12.4–15.9 (14.2 ± 1.0)	12.7–19.3 (14.3 ± 1.8)	11.8–12.9 (12.2 ± 0.4)	13.8	13.0–13.8 (13.4 ± 0.3)	
HL	5.7–7.6 (6.7 ± 0.5)	5.3–8.2 (6.8 ± 08)	5.5–6.1 (5.9 ± 0.3)	6.9	5.8–6.7 (6.2 ± 0.3)	
HW	7.0 –9.7 (8.2 ± 0.6)	6.8–10.7 (8.2 ± 1.1)	6.9–7.9 (7.4 ± 0.4)	8.1	7.6–8.4 (8.0 ± 0.3)	
IOD	2.7–3.7 (3.1 ± 0.3)	2.6–4.0 (3.1 ± 0.4)	2.3–2.9 (2.7 ± 0.2)	3.3	2.4–3.2 (2.9 ± 0.2)	
ED	2.2 –3.2 (2.8 ± 0.3)	2.3–3.8 (2.9 ± 0.4)	2.4–2.6 (2.5 ± 0.1)	2.9	2.1–2.8 (2.5 ± 0.2)	
TD	0.7–1.2 (1.0 ± 0.1)	0.4–1.3 (0.9 ± 0.2)	0.5–0.8 (0.7 ± 0.1)	0.9	0.4–0.9 (0.7 ± 0.2)	
AL	4.9–7.0 (5.8 ± 0.5)	5.1–7.1 (5.9 ± 0.7)	4.5–5.5 (5.0 ± 0.4)	5.6	4.3–5.7 (5.0 ± 0.4)	
HAL	8.5–11.8 (9.9 ± 0.9)	8.2–13.9 (9.8 ± 1.5)	8.5–9.9 (9.3 ± 0.4)	10.0	9.5–10.1 (9.7 ± 0.2)	
FI	3.8–5.9 (4.8 ± 0.6)	3.5–7.1 (4.8 ± 0.9)	3.6–4.4 (4.0 ± 0.3)	4.8	4.2–5.4 (4.9 ± 0.4)	
FII	5.4–7.5 (6.5 ± 0.6)	4.6–8.1 (6.1 ± 0.9)	5.3–6.2 (5.7 ± 0.3)	6.1	5.8–6.2 (6.0 ± 0.2)	
FIII	1.6–3.0 (1.9 ± 0.4)	1.1–2.2 (1.8 ± 0.3)	1.5–1.9 (1.6 ± 0.2)	1.8	0.8–1.7 (1.3 ± 0.3)	
TIII	1.3–1.9 (1.5 ± 0.3)	1.0–2.0 (1.5 ± 0.3)	1.4–1.6 (1.5 ± 0.1)	1.6	1.1–1.6 (1.3 ± 0.1)	
IND	1.8–2.8 (2.2 ± 0.2)	1.6–3.0 (2.2 ± 0.4)	1.6–2.1 (1.8 ± 0.1)	2.3	1.7–2.2 (2.0 ± 0.1)	
END	1.6–2.2 (1.9 ± 0.2)	1.5–2.2 (1.9 ± 0.3)	1.5–1.8 (1.6 ± 0.1)	1.9	1.7–2.0 (1.8 ± 0.1)	

Figure 4 Scatterplot of the discriminant analyses (DAPC) on the morphometric dataset of Centrolene buckleyi, C. elisae sp. nov. and C. marcoreyesi sp. nov.

The figure shows the two first axes on the morphometric data (from a total of 12 axes, which retain 96% of the variance).

Table 4 Osteological differences among Centrolene buckleyi sensu stricto and new species.

Species	Occipital	Squamosal	Parasphenoides	Vertebral profile Column	Humeral spine (adult males)	
Centrolene buckleyi sensu stricto	Condyles broad, projected, reaching level of exoccipital	Zygomatic ramus short, thick and rounded	Cultriform process with blunt anterior border, reaching level of neopalatine	Sacrum > III > IV > II > VI ≅ VII >VIII > V > I.	Longer than broad, equivalent to 40–44% of humerus length	
Centrolene elisae sp. nov.	Condyles thin, not projected, not reaching level of exoccipital posteriorly	Zygomatic ramus long and sub acuminated	Cultriform process with subacuminated anterior border, not reaching level of neopalatine	Sacrum > III > IV > II > VI ≅ VII ≅VIII > V > I.	Well-developed, curved, and broad, equivalent to 40–44% of humerus length	
Centrolene marcoreyesi sp. nov.	Condyles thin, slightly projected, not reaching level of exoccipital	Zygomatic ramus short and clawed shape	Cultriform process with rounded anterior border, not reaching level of neopalatine	Sacrum > III > IV ≅ V ≅ VI ≅VII ≅ VIII > II > I	Small and thin, equivalent to 25–30% of humerus length	

Table 5 Quantitative description of the advertisement calls (mean ± SD, range and n).

Parameters	Centrolene buckleyi
sensu stricto	Centrolene elisae
sp. nov.	Centrolene marcoreyesi
sp. nov.	
Number of recorded individuals	2	2	5	
Call duration (s)	0.261 ± 0.03 (0.192–0.297)
n = 16	0.203 ± 0.04 (0.156–0.229)
n = 13	0.085 ± 0.03 (0.053–0.131)
n = 35	
Inter-call interval (s)	35.4 ± 17.1 (18.7–71.3)
n = 10	13.7–15.0
n = 2	34.1 ± 15.0 (15.2–67.5)
n = 20	
Call rate (/min)	1.4–2.8
n = 2	4.1
n = 1	1.8 ± 0.2 (1.4–2.1)
n = 5	
Notes/call	1	2	1	
First note duration (s)	–	0.102 ± 0.02 (0.073–0.118)
n = 3	–	
Second note duration (s)	–	0.045 ± 0.009 (0.035–0.052)
n = 3	–	
Inter-note interval (s)	–	0.056 ± 0.008 (0.047–0.062)
n = 13	–	
Note rate (/s)	–	6.6 ± 1.5 (5.6–8.3)
n = 13	–	
Pulse/note	17.9 ± 1.5 (15–20)
n = 16	12.3 ± 4.0 (8–16)*
n = 13	8.1 ± 1.1 (6–10)
n = 35	
Pulse duration (ms)	10.7 ± 3.2 (4.7–22.2)
n = 284	7.9 ± 2.3 (4.8–14.2)*
n = 37	9.3 ± 2.4 (4.4–16.9)
n = 275	
Pulse rate (/s)	69.3 ± 6.7 (59.9–78.7)
n = 16	121.9 ± 12.1 (112.4–135.5)*
n = 13	103.6 ± 26.7 (58.8–137.5)
n = 35	
Dominant frequency (kHz)	2.9 ± 0.1 (2.8–3.2)
n = 16	3.7 ± 0.1 (3.6–3.8)
n = 13	3.3 ± 0.2 (2.8–3.4)
n = 35	
Frequency 5% (kHz)	2.8 ± 0.07 (2.8–3)
n = 16	3.5 ± 0.2 (3.4–3.7)
n = 13	3.0 ± 0.1 (2.6–3.2)
n = 35	
Frequency 95% (kHz)	3.3 ± 0.05 (3.2–3.4)
n = 16	3.8 ± 0.1 (3.7–3.9)
n = 13	3.4 ± 0.1 (3–3.6)
n = 35	
Number of visible harmonics	1–3	1–4	1–3	
Second harmonic frequency (kHz)	6.2 ± 0.2 (5.5–6.6)
n = 16	7.2 ± 0.3 (7–7.8)
n = 13	6.6 ± 0.2 (6.2–7.3)
n = 22	
Third harmonic frequency (kHz)	9.2 ± 0.3 (8.4–9.8)
n = 16	10.8 ± 0.4 (10.2–11.5)
n = 13	9.9 ± 0.3 (9.3–10.3)
n = 22	
Fourth harmonic frequency (kHz)	12.4 ± 0.4 (11.2–13.1)
n = 16	14.4 ± 0.7 (13.6–15.7)
n = 13	13.1 ± 0.4 (12.4–13.8)
n = 22	
Fifth harmonic frequency (kHz)	–	18.3 ± 0.8 (17.1–19)
n = 13	–	
Note:

Centrolene buckleyi sensu stricto (MZUTI 0763, KU 164507), Centrolene elisae sp. nov. (ZSFQ 5369, QCAZ 26032), and Centrolene marcoreyesi sp. nov. (KU 164511– KU 164513, MUTPL 271/FUTPL-A 140, MUTPL 272/FUTPL-A 141). *Values correspond to the first note of the call.

Color in life (MZUTI 763, ZSFQ 4420, DHMECN 13828). Dorsal surfaces bright to dark green, sharply demarcated laterally from lower white flanks; some individuals have scattered olive-green spots on the dorsum; throat and most of the venter pale green; parietal peritoneum yellowish-white; edge of upper lip white; ventrolateral borders of arms and tarsus white; small, white warts posterior to cloacal opening; bones green; gray–white iris with thin black reticulation and a horizontal brown stripe (Fig. 2).

Color in ethanol. Dorsal surfaces light to dark lavender, lower flanks white, ventral surfaces cream; ventrolateral borders of arms and tarsus white; upper lip white; parietal peritoneum white; all visceral peritoneum clear except for pericardium white.

Variation. Morphometric variation is shown in Table 3. Females larger than males; adult males with vocal slits, and dorsal skin with conspicuous spicules that are absent in females. Color variation is described in the “color in life” section.

Osteology. The following description is based on an adult male (MZUTI 0763). We present a detailed description of all skeletal elements.

Skull (Fig. 5A). Skull not ornamented or slightly ornamented on occipital, without exostosis or dermal modifications or co-ossification with skin. Maxillary arch complete; alary processes of premaxillae small and with pointed ends; maxilla broadest anteriorly, tapering posteriorly; pars facialis broad. Quadratojugal ossified and broad, overlapping anteriorly with maxilla and posteriorly articulated with ventral ramus of squamosal. Two ossified nasals, relatively small, separated from each other, posterolaterally articulated to neopalatine; nasals not articulating with sphenethmoid. Sphenethmoid forming the anterior part of braincase; anterior margin of bony sphenethmoid at level of plane antorbitale, and posterior margin at about anterior third of orbit. In dorsal view, sphenethmoid articulates posterolaterally with paired frontoparietals. Frontoparietals independent, not ornamented, arranged in parallel, posteriorly fused to the occipital. Frontoparietal fontanelle delimited by sphenethmoid anteriorly, frontoparietals laterally, and prootic posteriorly. Occipital fontanelles absent.

Figure 5 Cranial osteology of closely related Centrolene species; from left to right: dorsal, ventral, frontal, and lateral views.

(A) Centrolene buckleyi sensu stricto, male, MZUTI 0763; (B) C. elisae sp. nov., male holotype MZUTI-084; (C) C. marcoreyesi sp. nov., male holotype, ZSFQ 4418. Labels: AP, alary process of premaxilla; AS, angulosplenial; COL, columella; D, dental; EXO, exoccipital; FP, frontoparietal; MMK, mentomeckelian bone; MX, maxilla; NA, nasal; NPL, neopalatine; OC, occipital condyle; PM, premaxilla; PO, prootic; PS, parasphenoid; PT, pterygoid; QJ, quadratojugal; SQ, squamosal; SE, sphenethmoid; SM, septomaxilla; V, vomer. Prepared by Daniela Franco-Mena.

Prootics and exoccipitals co-ossified; crista parotica completely ossified. Neopalatine present and in contact with sphenethmoid. Maxillary and premaxillary teeth short and monocuspid. Suspensorium composed of paired pterygoids and squamosals; zygomatic ramus of squamosal short, thick and rounded anterior border, otic ramus slightly posterosuperior oriented. Each pterygoid consists of anterior, medial, and posterior rami. Anterior ramus articulating with posterior end of maxilla; medial ramus coverings the prootic pseudobasal process; posterior ramus oriented towards ventral ramus of the squamosal. Lower jaw composed of paired mentomeckelian bones and dentary. Moderately-sized vomers broadly separated from one another medially, each composed of arcuate bone bordering anterior and medial margins of choana. Prechoanal and postchonal rami thin and unexpanded distally. Slender dentigerous processes extending ventromedially from the union of the pre- and postchoanal processes. Neopalatines unornamented, arcuate, and articulating with lateral margin of sphenethmoid just anterior to the orbitonasal foramen. Neopalatines narrowly separated from maxilla. Parasphenoid large and broad, anterior end blunt, overlapping sphenethmoid, nearly reaching level of neopalatines; alar processes of parasphenoid relatively short and partially fused to occipital; short posteromedial process present, but distinctly separated from margin of foramen magnum. Columella present, thin. Pterygoid with three branches: anterior ramus curved, oriented anterolaterally toward the maxilla, with which it articulates at approximately midlength of orbit; medial and posterior rami of pterygoid about equal in length; medial ramus in contact with edge of ossified lateral margin of prootic.

Forelimb and hind limb (Fig. 6A). The forelimb is composed of humeral bone, radioulna, carpal elements, prepolex, and four digits (I–IV). The hind limb consists of a femur, tibiale, fibula, fibulare (=astragalus), metacarpals, metatarsals, and five digits (I–V). The phalangeal formulae for the hand and foot are standard 2-2-2-3 and 2-2-3-4-3, respectively. Order of finger length: I < II < IV < III, and in toes: I < II < V < III < IV. Metacarpals long and slender; distal end rounded; inner edge of Finger III with dilated medial metacarpal process (Hayes & Starrett, 1980). Prepollex well developed, broad and curved, with a rounded distal end. Intercalary element between the last phalanges of all digits; terminal phalanx with T- or Y-shaped end. Carpus is composed of Carpal 1, Element Y, and a large postaxial element assumed to represent a fusion of Carpals 2–4, radiale, and ulnar. Element Y seems to be partially fused with prepollex; prepollex composed of one small bone. Tarsus is composed of three tarsal elements, presumably Tarsal 1 + 2 + 3. Humeral bone with well-developed humeral spine (in males), equivalent to 40–44% of humerus length, oriented at an angle of 35–45° in relation to axis of humeral bone.

Figure 6 Post-cranial osteology of closely related Centrolene species; from left to right: dorsal, ventral, frontal, and lateral views.

(A) Dorsal view. (B) Ventral view. (C) Frontal view. (D) Lateral view. Labels: AP, alary process of premaxilla; AS, angulosplenial; COL, columella; D, dental; EXO, exoccipital; FP, frontoparietal; MMK, mentomeckelian bone; MX, maxilla; NA, nasal; NPL, neopalatine; OC, occipital condyle; PM, premaxilla; PO, prootic; PS, parasphenoid; PT, pterygoid; QJ, quadratojugal; SQ, squamosal; SE, sphenethmoid; SM, septomaxilla; V, vomer. Prepared by Daniela Franco-Mena.

Pectoral girdle (Fig. 6A). The pectoral girdle is composed of scapula, suprascapula, zonal area (coracoid, cleithrum, and clavicle) and posteromedial process. Suprascapula completely mineralized, with cleithrum apparent as a slender bone along its leading edge; cleithrum ossified. Clavicles oriented anteromedially, with the medial tips distinctly separated from one another; anterolateral end of the clavicle articulating with scapula.

Vertebral column and pelvic girdle (Fig. 6A). Vertebral column with eight presacral vertebrae; presacrals I and II notably shorter than posterior presacral. All presacrals are non-imbricate except the first, which is partially imbricate. Neural arch of Presacral II bearing a rounded, medial process that articulates with neural arch of Presacral I. Vertebral profile in decreasing order of overall width of bony parts sacrum > III > IV > II > VI ≅ VII >VIII > V > I. Orientations of transverse processes of Presacrals II, VII, and VIII directed anterolaterally, and those of Presacrals III, IV, V, and VI with clear posterolateral orientation. Sacral diapophyses moderately dilated laterally; leading edge and posterior margin of diapophyses slightly concave. Urostyle long and slender, with bicondylar articulation with the sacrum, and bearing a low dorsal crest throughout its anterior half. Length of urostyle less than combined length of presacral vertebrae. Pelvic girdle composed of ischium, ilium, and pubis. Ilial shafts cylindrical, lacking dorsal crest. Ilia tightly joined with ischia and pubes. Pubis ossified.

Distribution. Centrolene buckleyi sensu stricto is distributed along the northern to the central portion of the Cordillera Oriental and Cordillera Occidental of the Andes in Ecuador (Fig. 7) and inhabits Western Montane Forest, Andean Shrub, Páramo, Eastern Montane Forest ecoregions (Ministerio del Ambiente del Ecuador (MAE), 2012), and pasture at elevations between 2,677–3,416 m (Guayasamin et al., 2020; this study). As mentioned above, the neotype of C. buckleyi sensu stricto is from northern Ecuador (Isla Wolf in Laguna de Cuicocha, 3,070 m, Imbabura; Guayasamin et al., 2020). The individuals reported from Guarumales by Guayasamin et al. (2020) correspond to C. marcoreyesi sp. nov. (described below). Given the phylogenetic and acoustic results of our study, we consider the presence of C. buckleyi sensu stricto as uncertain in Colombia and Peru.

Figure 7 Phylogeny and distribution of Centrolene, highlighting lineages closely related to Centrolene buckleyi, including the two new species described herein.

Collapsed clades and outgroups not shown (see Fig. 1 for complete tree). Circles = localities of specimens used for the phylogenetic analyses; Triangle = occurrence of C. lemniscata, a species morphologically similar to C. buckleyi, but for which there are no molecular sequences available. In the phylogenetic tree the highlighted species correspond to colors of circles on the map. Prepared by Mateo A. Vega-Yánez.

Call. In the literature there are three descriptions of the advertisement call of Centrolene buckleyi (Bolívar, Grant & Osorio, 1999; Guayasamin et al., 2006a, 2020). However, from these, only the call recorded nearby Oyacachi, Napo province (MZUTI 0763; Guayasamin et al., 2020) can be attributed to C. buckleyi sensu stricto. For the description of the advertisement call, we analyzed this last recording of the specimen MZUTI 0763, from Oyacachi (0.2189°S, 78.044°W; 3,012 m), Napo province, Ecuador, recorded by Italo Tapia on 17 May 2012, at 01:02 h, 10 °C, and KU 164507, from 11 km E Papallacta (2,660 m), Napo province, Ecuador, recorded by William E. Duellman on 22 March 1975, at 22:30 h, 10 °C.

Centrolene buckleyi sensu stricto has an advertisement call composed by a pulsed, single “Tri” type (sensu Duarte-Marín et al., 2022) note (Fig. 8A). Descriptive statistics of the acoustic variables are provided in Table 5. The calls are characterized by a mean duration of 0.261 s, a mean inter-call interval of 35.4 s, and a call rate of 1.4–2.8 calls/min. Each call (note) consists on average of 17.9 pulses, with a mean pulse duration of 10.7 ms, and a mean pulse rate of 69.3 pulses/s. The mean dominant frequency of the notes is 2.9 kHz, with a mean 90% bandwidth of 2.8–3.3 kHz (Table 5). The call exhibits a slight ascending frequency modulation, with an initial dominant frequency at 2.8–3 kHz (2.9 ± 0.02, n = 16) that increases to 3.1–3.4 kHz (3.2 ± 0.08, n = 16) final dominant frequency. Three harmonics are visible (Fig. 8A). The male MZUTI 0763 also emitted a series of five consecutive calls. These calls have the same duration and structure as the regular calls, but the inter-call interval was considerably shorter: 0.7–0.9 s (0.8 ± 0.09, n = 4). It is very probable that this type of vocalization was triggered by the presence of nearby females or competitive males, as documented in many anuran species (see Wells, 2007 for a detailed discussion).

Figure 8 Oscillograms and spectrograms of the advertisement calls of three species of Centrolene with a multipulsed “Tri” type structure.

(A) Centrolene buckleyi sensu stricto (MZUTI 763), (B) Centrolene elisae sp. nov. (ZSFQ 5369), (C) Centrolene marcoreyesi sp. nov. (MUTPL 271/FUTPL-A 140). Note that the call of C. elisae sp. nov. is composed by two defined notes. Prepared by Diego Batallas-Revelo.

Generic placement of the new species. The two new species are placed in the genus Centrolene Jiménez de la Espada, 1872, based on molecular phylogenetics (Fig. 1) and morphological data (see below). All species in Centrolene (sensu Guayasamin et al., 2009) share the following traits: (1) humeral spines present in adult males (except Centrolene daidalea Ruiz-Carranza & Lynch (1991) and C. savagei Ruiz-Carranza & Lynch, 1991); (2) tri-, tetra-, or pentalobed liver, covered by a transparent hepatic peritoneum; (3) ventral parietal peritoneum translucent posteriorly and white anteriorly; (4) bones varying from green to pale gray in life; and (5) nuptial pads conspicuous in adult males. The two new species described herein presents all the aforementioned traits and its placement within Centrolene is unambiguous.

Centrolene elisae sp. nov. Daniela Franco-Mena, Mateo A. Vega-Yánez, Juan Pablo Reyes-Puig, Juan M. Guayasamin

LSID: E9E154BD-3D98-471B-8A41-7F8870CA1572

Centrolene buckleyi—Guayasamin et al. (2006a)

Centrolene buckleyi [Ca2]—Amador et al. (2018)

English common name. Elisa’s Glassfrog

Spanish common name. Rana de Cristal de Elisa

Holotype. MZUTI 84 (Fig. 3), adult male, from Las Caucheras (0.6133°S, 77.8974°W; 2,187–2,191 m), Napo province, Ecuador, collected by Gisela Bragado and Henry Grifo on 26 August, 2011.

Paratypes. (one female, five males). MZUTI 0083 and MZUTI 0085, adult males, same data as holotype. ZSFQ 4228 (Fig. 2) adult male, from Chamanapamba Reserve (1.4237°S, 78.3932°W; 2,586 m), Tungurahua province, Ecuador, collected by Daniela Franco-Mena, Tasman Rosenfeld, David Brito-Zapata, and Tito Recalde on 23 June, 2021. DHMECN 4800, adult male, from Río Pucayacu, eastern flank of Tungurahua volcano (1.436245°S, 78.409335°W; 2,400 msnm), Tungurahua province, Ecuador, Collected by Juan Pablo Reyes-Puig and Nelson Palacios on 28 April, 2007. ZSFQ 5367 an adult female, ZSFQ 5368, and ZSFQ 5369 adult males (Fig. 2), from Yanayacu Biological Reserve (0.61424°S, 77.8821°W; 2,118 m), Napo province, Ecuador collected by Daniela Franco-Mena, José Simbaña, Katherine Stroh, and Mateo A. Vega-Yánez on 14 April, 2023.

Definition. (1) SVL in adult males 22.9–25.3 mm (n = 6), in an adult female 27.2 mm; (2) in life, dorsum green, usually with minute whitish spots; anterior half of venter whitish, posterior half translucent; (3) iris gray-white with thin black-brown reticulation and a horizontal brown stripe; (4) humeral spines and vocal slits present in adult males; (5) snout rounded in dorsal profile, inclined in lateral profile; (6) webbing absent between Fingers I and II, webbing basal between II and III, outer fingers III (21/2–22/3)–(21/2–2+) IV; (7) webbing on foot: I (11/2–2−)–(2+–21/3) II (1+–11/3)–(2+–22/3) III (1–11/2)–(21/3–22/3) IV (22/3–3−)–(11/2–2) V; (8) ulnar fold low; inner tarsal fold short; outer tarsal fold low; (9) prepollex concealed; nuptial excrescences present, Type-I; (10) Toe I shorter than Toe II.

Comparison with similar species. Centrolene elisae is differentiated from its congeners mainly by having a dark green dorsum with minute whitish spots, white upper lip, inclined snout, rounded in dorsal profile, relatively medium-sized humeral spine (in adult males), and reduced webbing between inner fingers (Fig. 3). Differences between the new species and morphologically similar taxa (i.e., C. buckleyi sensu stricto, C. venezuelense, C. marcoreyesi sp. nov.) are summarized in Table 2. Skull key characters are summarized in Table 4. Genetic distances are available in Table S2 and Fig. S1. Additionally, Centrolene elisae has a well-defined two-note advertisement call that structurally differs from the one-note calls exhibited by closely related species (Fig. 8 and Table 5; see the Call section below).

Description of the holotype. Adult male, MZUTI 0084 (Fig. 2); moderate size (SVL = 24, 5 mm). Snout rounded in dorsal profile, sloping in lateral profile; upper lip white, loreal region slightly concave; internarial area barely depressed. Eye small (ED 10% of SVL), directed anterolaterally. Tympanic annulus indistinct in its upper portion; tympanic membrane differentiated from skin around the tympanum. Dentigerous processes of vomers lacking teeth; tongue ovoid; vocal slits extending posterolaterally from base of tongue to angle of jaws.

Medium-sized humeral spine present, curved. Webbing absent between fingers I and II, webbing basal between II and III, outer fingers III 22/3–2+ IV; disc on third finger larger than those on toes, and shorter than eye diameter, finger discs truncate; subarticular tubercles rounded, and flat, abundant supernumerary tubercles present over a granular palm; palmar tubercle large, elliptical; thenar tubercle indistinct. Legs slender; heels of adpressed limbs perpendicular to body slightly overlap. Length of tibia 59,4% of SVL; inner metatarsal tubercle large, flat, elliptical; outer metatarsal tubercle indistinct. Subarticular tubercles rounded and flat; supernumerary tubercles present over the granular palm. Webbing on foot: I 11/2–2+ II 1+–2−2/3 III 1+1/–22/3 IV 3− –13/4 V; disc on Toe I slightly expanded, all other discs rounded to fairly truncate, pointed papillae on discs absent. Skin on dorsal surfaces of head, body, and lateral surface of head and flanks shagreen, covered with minute spinules and spots; throat smooth; venter and lower flanks areolate; cloacal opening directed posteriorly at the upper level of thighs; subcloacal area granular.

Measurements (in mm) of the holotype (MZUTI 0084). SVL = 24.5, FEL = 13.2, TL = 14.6, FL = 12.0, HL = 5.6, HW = 7.0, IOD = 2.8, ED = 2.5, TD = 0.7, AL = 5.2, HAL = 8.5, FI = 3.9, FII = 5.4, FII = 1.5, TIII = 1.4, IND = 1.8, END = 1.5.

Measurements (in mm) of type series. Morphometric variation of the type series is summarized in Table 3.

Color in life. Dorsal surfaces dark green with small to minute white spots; upper flanks sharply demarcated laterally from lower white flanks; throat and most of the venter pale green; parietal peritoneum yellowish white; whitish-yellow labial line present; ventrolateral borders of arms and tarsus white; small, white spots posterior to cloacal opening corresponding to pericloacal warts; bones green; copper–white and gray iris with thin black reticulation and a horizontal brown stripe. Digits and disks green and yellowish interdigital webbing (ZSFQ 5367, ZSFQ 5369, ZSFQ 4428; Fig. 2).

Color in ethanol. Dorsal surfaces of body lavender to grayish lavender with few to numerous minute white dots; white upper lip. Dorsal surfaces of limbs cream to light lavender, with or without minute cream spots. Pericardium white, other visceral peritoneum clear. Cloacal ornamentation and ulnar and tarsal folds with a thin layer of iridophores. Melanophores present from dorsal surfaces of fingers; toes with melanophores restricted to Toe V or, rarely, Toe IV (Fig. 3).

Variation. Morphometric variation is shown in Table 3 and Fig. 4. The only known female is larger than the males. One male (ZSFQ 4428) had a slightly darker dorsal coloration, and more dorsal spicules than other individuals (Fig. 2).

Osteology (MZUTI 0084). To minimize redundancy in the osteology descriptions, we focus on distinctive osteological features, noting differences from other species (Table 4 and Figs. 5B, 6B). Other traits are as those described for Centrolene buckleyi (see above).

Natural history. Centrolene elisae, as other glassfrogs, is a nocturnal species found on vegetation along streams. At Las Caucheras (Figs. 9A and 9B) six individuals were found on the leaves of shrubs and ferns in a paddock near a small stream and a river, approximately 20 to 250 cm above ground level; five individuals (MZUTI 83–85, ZSFQ 5368, 69) were calling. At Río Pucayacu, a calling male (DHMECN 4800) was found on vegetation 2 m above stream level. At Chamanapamba Reserve (Figs. 9C and 9D), one individual (ZSFQ 4228) was calling, perched on a fern leaf 230 cm above ground level, near a small stream. The streams where the species was recorded were between 100–150 cm wide, in primary and secondary forest. At Yanayacu Biological Station, intensive inventories for 3 years (2002–2004) resulted in only three individuals of C. elisae suggesting that this species is rare (Guayasamin et al., 2006a); also near Yanayacu reserve, the area for cattle ranching and agriculture continues growing, reducing the habitat of the species. At Chamanapamba reserve C. elisae is syntopic with Nymphargus sp., Hyloscirtus sethmacfarlanei, Pristimantis donnelsoni, and P. marcoreyesi sp. nov. (DFM and JRP pers. comm.; Reyes-Puig et al., 2022a).

Figure 9 Habitat of Centrolene elisae sp. nov. and C. marcoreyesi sp. nov.

(A and B) Yanayacu Biological Station, Napo province, (C and D) Chamanapamba reserve, Tungurahua province, (E and F) Abra de Zamora (MUTPL 271, 272), and (G) Guarumales (CJ 12631, CJ 11564, CJ 11364), Zamora Chinchipe province. Photographs (A) by Mateo Vega-Yánez, (B, D) by Daniela Franco-Mena, (C) by Juan Pablo Reyes-Puig, (E, F) by Paul Székely, and (G) by Jaime Culebras.

Eggs. At Chamanapamba reserve we found three egg clutches with embryos at Gosner Stage 22 (Figs. 10A and 10B). The egg clutches were attached to the upper side of a leaf at ~170 cm above the small stream. The first clutch contained 28 embryos, the second clutch contained 21, and the third clutch contained 47 embryos; an adult male (ZSFQ 4428) was observed near the eggs.

Figure 10 Egg-clutches of Centrolene elisae sp. nov. from Chamanapamba reserve (A and B) and of Centrolene marcoreyesi sp. nov. from Guarumales (C and D).

Photographs (A, B) by Daniela Franco-Mena and (C, D) by Jaime Culebras.

Call. For the description of the advertisement call, we analyzed one recording (ZSFQ 5369), from Yanayacu Biological Station (0.6150°S, 77.88189°W; 2,118 m), Napo province, Ecuador, made by Daniela Franco-Mena on 11 March 2023, at 23:19 h, 10 °C. Centrolene elisae has a “Tri” type advertisement call, composed by two pulsed notes (Fig. 8B). Descriptive statistics of the acoustic variables are provided in Table 5. The calls are characterized by a mean duration of 0.203 s, an inter-call interval of 13.7–15.0 s, and a call rate of 4.1 calls/min. The first note is much longer than the second one and has a mean duration of 0.102 s, consists on average of 12.3 pulses, with a mean pulse duration of 7.9 ms, and a mean pulse rate of 121.9 pulses/s; the second note has a mean duration of 0.045 s and consists of 2 or 3 pulses. The mean inter-note interval is 0.056 s and the note rate is about 6.6 notes/s. The mean dominant frequency of the calls is 3.7 kHz, with a mean 90% bandwidth of 3.5–3.8 kHz (Table 5). The first, longer note, exhibits a very slight ascending frequency modulation, with an initial dominant frequency at 3.4–3.8 kHz (3.6 ± 0.2, n = 3) that increases to 3.6–3.9 kHz (3.7 ± 0.1, n = 3) final dominant frequency. The fundamental frequency is not recognizable, but 4 harmonics are visible (Fig. 8B).

We also accessed calls from another individual (QCAZ 26032) obtained in laboratory conditions (see Guayasamin et al., 2006a); the recordings of this male are not good enough to allow precise measurements of the emitted notes and pulses. Nevertheless, we were able to measure the frequencies, call duration and inter-call interval. These recordings are interesting because document the vocalization of interacting males; the calls have the same structure, being composed by two notes, but are different from the regular advertisement call mostly by the much shorter inter-call interval (the calls were emitted more frequently). The calls have a slightly longer duration of 0.198–1.284 s (0.264 ± 0.03, n = 8), but much shorter inter-call interval of 0.343–0.844 s (0.454 ± 0.2, n = 6), and a much higher call rate of 81.4–89.6 calls/min. The dominant frequency of the notes of 3.5–4.1 kHz (3.8 ± 0.3, n = 8) and the 90% bandwidth ranging from 3.4–3.5 kHz (3.5 ± 0.05, n = 8) to 4–4.2 kHz (4.1 ± 0.08, n = 8) were very similar to the frequencies of the regular advertisement calls (Table 5).

The advertisement call of C. elisae is different from the call of C. buckleyi sensu stricto and C. marcoreyesi sp. nov. (calls can be distinguished even by ear) as they have a distinctive structure (C. elisae has a call composed of two notes, whereas C. buckleyi and C. marcoreyesi sp. nov. have only one note per call) and non-overlapping dominant frequencies (Fig. 8 and Table 5). A somewhat similar, double noted call (but with longer call duration of about 0.668 s and lower dominant frequency of about 2.8 kHz) was described also in C. condor, but the call of C. condor has a much lower dominant frequency (2.6–3.0 KHz; Almendáriz & Batallas, 2012) when compared to C. elisae (3.6–3.8 KHz). The call of C. elisae is also different from the advertisement call of its sister species, C. venezuelense (Rivero, 1968), as this species has a call usually composed by a series of four notes (much shorter than the first note of the C. elisae call) and higher dominant frequency of about 3.9–4.4 kHz (Señaris & Ayarzagüena, 2005). Finally, it seems that the advertisement call of C. elisae is also different from the call of C. cf. venezuelense from Colombia, which has a call composed of only one, longer pulsed note more similar to the call of C. buckleyi sensu stricto (KVV personal observation).

Distribution. Centrolene elisae is endemic to the cloud forests of the eastern Cordillera of the Ecuadorian Andes (Fig. 7). The species has been documented from four localities: Las Caucheras, Yanayacu Biological Station (Napo Province), Chamanapamba Reserve, and Río Pucayaku in Nelson Palacios Reserve (Tungurahua Province), at elevations of 2,100–2,586 m.

Conservation status. Centrolene elisae is know from only four localities on the Amazonian slopes of the Andes (Fig. 7), an area that has suffered deforestation and fragmentation because of agriculture and cattle farming (Gaglio et al., 2017). The current estimated extent of occurrence for the species is <5,000 km2. Therefore, following IUCN criteria to assess the current extinction risk of the species (Gärdenfors et al., 2001), we propose that C. elisae should be considered as Endangered (EN) B1. b (ii) c.

Phylogenetics. Centrolene elisae is inferred monophyletic, with significant BI support (posterior probability = 1) and moderate ML support (bootstrap = 84). The new species forms part of a clade with an unresolved polytomy with C. venezuelense, a potential new species from Colombia (C. cf. venezuelense; IAvH-Am-17401, 17403, 17407, 17410), and another one from northern Ecuador (Centrolene sp.; ZSFQ 2134) (Fig.1).

Etymology. The species epithet “elisae” is a noun in genitive case, with the Latin suffix “e” (ICZN 31.1.2). We are pleased to dedicate the species to Elisa Bonaccorso (Fig. S2), in recognition for her contributions to bird systematics and biogeography (Bonaccorso, 2009; Bonaccorso, Koch & Peterson, 2006; Bonaccorso et al., 2011; Sornoza-Molina et al., 2018), conservation biology (Lessmann, Munoz & Bonaccorso, 2014; Lessmann et al., 2016; Bonaccorso et al., 2021), batrachology (Bonaccorso et al., 2003; Guayasamin & Bonaccorso, 2004), and her passionate commitment to the education of the next generation of scientists. This is also a recognition for her luminous presence in my life (Juan Manuel Guayasamin).

Centrolene marcoreyesi sp. nov. Daniela Franco-Mena, Paul Székely, Jaime Culebras, Diego Batallas-Revelo, Juan Pablo Reyes-Puig, Juan M. Guayasamin

LSID: FC59A40D-6F9C-45BF-A6FE-280FA20BC2D1

Centrolene buckleyi [Ca1]—Amador et al. (2018)

English common name. Marco Reyes´ Glassfrog

Spanish common name. Rana de Cristal de Marco Reyes

Holotype. ZSFQ 4418 (Fig. 3), adult male from Estación Científica San Francisco (3.971667°S, 79.079167°W; 1,840 m), Zamora Chinchipe province, Ecuador, collected by Marco Reyes-Puig and Sebastián Valverde on 15 February, 2012.

Paratypes. ZSFQ 4417 adult male, same data as the holotype; MUTPL 271, 272, adult males, from Abra de Zamora (3.9689°S, 79.1110°W; 2,190 m) (Fig. 2), Zamora Chinchipe province, Ecuador, collected by Paul Székely and Diana Székely on 29 April, 2017; CJ 11364, adult male, from Guarumales (3.94049°S, 78.986891°W; 2,070 m), Zamora Chinchipe province, Ecuador, collected by Jaime Culebras, Santiago Hualpa, Daniel Hualpa, and Darwin Núñez on 27 February, 2020; CJ 11564, adult male, from Guarumales (3.93491°S, 78.99956°W; 2,008 m), Zamora Chinchipe province, Ecuador, collected by Jaime Culebras and Darwin Nuñez on 22 February, 2021; CJ 12631, adult male from Guarumales (3.93825°S, 79.00525°W; 2,109 m), Zamora Chinchipe province, Ecuador, collected by Jaime Culebras, Daniel Hualpa and Santiago Hualpa on 15 April 2022.

Definition. Within Centrolene, Centrolene marcoreyesi sp. nov. is defined by the following set of traits: (1) SVL in adult males 24.5–27.0 mm (n = 6), unknown in females; (2) in life, dorsum shagreen usually with low whitish spots; anterior two-thirds of venter whitish, posterior third translucent; (3) in life, iris white-lavender with fine brown reticulations; (4) humeral spines, vocal sac and slits present in adult males; (5) snout rounded in dorsal profile, sloping in lateral profile; (6) webbing absent between inner finger and Finger II, reduced to moderate between outer fingers: III (21/3–23/4)–(21/4–22/3) IV; (7) webbing on feet: I (11/2–12/3)–(2––2) II (1+–11/4)–(21/4–2+) III (11/2–12/3)–(21/3–21/2) IV (2+–21/2)–(12/3–3–) V; (8) inner and outer ulnar and tarsal folds conspicuous; (9) concealed prepollex; nuptial excrescences present, Type-I; (10) Toe I shorter than Toe II.

Comparison with similar species. Centrolene marcoreyesi is differentiated from its congeners by having dorsal skin shagreen with light dispersed low warts, yellowish-white upper lip, sloping snout in lateral profile, relatively small humeral spine (in adult males), and reduced webbing between inner fingers (Fig. 3). Differences between the new species and morphologically similar taxa (i.e., C. buckleyi sensu stricto, C. venezuelense, C. elisae) are summarized in Table 2. Skull key characters are summarized in Table 4. Genetic distances are available in Table S2 and Fig. S1.

Description of holotype. Adult male, ZSFQ 4418, of moderate size (SVL = 25.9 mm) (Fig. 3). Snout rounded in dorsal profile, sloping in lateral profile; upper lip white, loreal region slightly concave; internarial area barely depressed. Eye small (eye diameter = 10% of SVL), directed anterolaterally. Tympanic annulus differentiated, but obscured in its upper portion by the supratympanic fold; tympanic membrane differentiated, clearly thinner than skin found around the tympanum. Dentigerous processes of vomers lacking teeth; tongue ovoid, with notched posterior border; vocal slits extending posterolaterally from the base of the tongue to angle of jaws. Humeral spine present, relatively small, curved, and pointy at its distal end. Webbing absent between Fingers I–III, reduced between outer fingers: III 22/3—21/2 IV; disc on third finger larger than those on toes, and smaller than eye diameter; finger discs truncate; subarticular tubercles rounded; abundant supernumerary tubercles on palm; palmar tubercle large, elliptical; thenar tubercle indistinct. Legs slender; heels overlapping when adpressed perpendicularly to the body. Length of tibia 59.8% of SVL; inner metatarsal tubercle large, flat, elliptical; outer metatarsal tubercle indistinct. Subarticular tubercles rounded and flat; numerous supernumerary tubercles on granular palms. Webbing on feet: I 12/3–2+ II 11/4–21/4 III (11/2–12/3)–(21/3–21/2) IV (2+–21/2)–(12/3–3–) V; all disc toes slightly expanded; discs lacking pointed projections (papillae). Inner and outer ulnar and tarsal folds conspicuous.

Measurements (in mm) of the holotype (ZSFQ 4418). SVL = 25.9, FEL = 13.7, TL = 15.5, FL = 13.8, HL = 6.2, HW = 7.6, IOD = 2.9, ED = 2.7, TD = 0.9, AL = 5.1, HAL = 9.7, F1 = 5.3, FII length = 6.2, FIII = 1.7, TIII = 1.2, IND = 2.1, END = 1.8.

Measurements (in mm) of the type series. Morphometric variation of the type series is summarized in Table 3.

Color in life. Description based on color photographs of MUTPL 271 (Fig. 2). Dorsal surfaces of body, arms, and limbs green with numerous whitish spots of various sizes. Yellowish-white upper lip; anterior two-thirds of venter yellowish-white, posterior third translucent. Fingers, toes, and membranes yellowish-green. Bones green. Iris white with a slight lavender tone, with fine brown reticulations.

Color in ethanol. Dorsum lavender, with yellowish-white dots distributed along the dorsum. Some individuals (CJ 11364, 11564, 12631) present a grayish-lavender dorsum with white spots; white upper lip. Anterior one third to two-thirds of the parietal peritonium white, hepatic peritoneal translucid venter yellowish-cream, posterior third translucent (Fig. 3).

Variation. Morphometric variation is shown in Table 3 and Fig. 4. One male (ZSFQ 4428) had a slightly darker dorsal coloration, and more dorsal spicules than other individuals (Fig. 2). Individuals from the type locality in the Estación Científica San Francisco (ZSFQ 4417, 18) exhibit a more reduced webbing between Fingers III and IV than the rest of the specimens represented in the type series.

Osteology (ZSFQ 4418). To minimize redundancy in the osteology description (skull, forelimb, hind limb, pectoral girdle, vertebral column, and pelvic girdle) we focus on distinctive osteological features, noting differences from other species in Table 4, Figs. 5C and 6C. We provided a detailed description of the osteology in Centrolene buckleyi (see in the section of osteology).

Natural history. The holotype ZSFQ 4418 was collected at night in a small stream, on herbaceous vegetation. In Abra de Zamora (Figs. 9E and 9F), several individuals were calling near small streams in an evergreen upper montane forest ecosystem (Homeier et al., 2008). The encountered males were calling from the upper surfaces of leaves, at about 1 m from the water surface. In Parque Nacional Podocarpus, several males were observed calling from about 2 m on leaves over fast flowing streams. At Guarumales (Fig. 9G) four individuals (CJ 10139, 10140, 10158, 10305) were found calling near a river, approximately 150 to 350 cm above ground level; two individuals (CJ 11366, 11372) were found on leaves, about 100–200 cm above ground level; the two other individuals were observed on fern leaves. A male (CJ 12631) was observed calling nearby a clutch with 19 eggs (one of them dead; Figs. 10C and 10D). A male (CJ 11564) was observed calling from a fern leaf at 150 cm above ground level; the streams where the species is present are variable in width, between 2 to 8 m. Males were found in primary and secondary forests and on the edge of pastures. At Abra de Zamora C. marcoreyesi was sympatric with Gastrotheca testudinea and in Guarumales with Nymphargus cariticommatus, N. posadae, N. cochranae, and Hyalinobatrachium sp.

Call. For the description of the advertisement call, we analyzed three recordings (KU 164511– KU 164513) made by William E. Duellman on 8 March 1975 in Abra de Zamora, Zamora Chinchipe province, Ecuador, between 21:45 and 21:55 h, at 11 °C and two recordings (MUTPL 271/FUTPL-A 140 and MUTPL 272/FUTPL-A 141), from Abra de Zamora (3. 9689°S, 79.1110°W; 2,190 m), Zamora Chinchipe province, Ecuador, recorded by Paul Székely on 29 April 2017, between 20:49 and 21:08 h, at 14.5 °C.

Centrolene marcoreyesi has a “Tri” type of advertisement call composed by one pulsed note (Fig. 8C). Descriptive statistics of the acoustic variables are provided in Table 5. The calls are characterized by a mean duration of 0.085 s, a mean inter-call interval of 34.1 s, and a mean call rate of 1.8 calls/min. Each call (note) consists on average of 8.1 pulses, with a mean pulse duration of 9.3 ms, and a mean pulse rate of 103.6 pulses/s. The mean dominant frequency of the notes is 3.3 kHz, with a mean 90% bandwidth of 3.0–3.4 kHz (Table 5). The call exhibits a slight ascending frequency modulation, with an initial dominant frequency at 2.8–3.3 kHz (3.1 ± 0.2, n = 35) that increases to 3–3.6 kHz (3.3 ± 0.2, n = 35) final dominant frequency. The fundamental frequency is not recognizable, but up to 3 harmonics are visible (Fig. 8C). The two males MUTPL 271 and 272, recorded in 2017, emitted mostly two consecutive calls, besides the regular ones. These notes have the same duration, structure and frequency as the regular calls, but the inter-call interval was significantly shorter, of 0.8–1.7 s (1.1 ± 0.3, n = 10). It is almost certain that this behavior was triggered by the nearby presence of a competitive male as the distance between the two recorded individuals was smaller than half a meter.

It is worth mentioning that we observed some differences between the calls recorded, in the same locality, in 1975 and the ones from 2017. Thus, the calls from 1975 had a longer call duration of 0.098–0.131 s (0.116 ± 0.01, n = 13), lower pulse rate of 58.8–93.4 pulses/s (70.9 ± 10.0, n = 13), and slightly lower dominant frequency of 3–3.4 kHz (3.3 ± 0.1, n = 13) than the calls from 2017, with a call duration of 0.053–0.081 s (0.067 ± 0.01, n = 22), a pulse rate of 103.3–137.5 pulses/s (122.9 ± 6.5, n = 22), and dominant frequency of 3.5–3.6 kHz (3.5 ± 0.02, n = 22). These differences where probably caused by the different recording conditions (especially the temperature, as in 2017 it was higher) and/or a combination of general weather conditions and calling behavior of the males. We also have to note that probably the specified altitude of the recordings (made on the tape as 2,850 m) is probably incorrect because according to the available data C. marcoreyesi (identified in 1975 by Duellman as C. buckleyi) is limited to an altitude of about 2,200 m. The altitude of the recording corresponds to the higher part of Abra de Zamora, the crest between Loja and Zamora Chinchipe provinces, from where there are no records of glassfrogs (it is a subpáramo ecosystem).

The advertisement call of C. marcoreyesi has different structure from C. elisae (one note per call in C. marcoreyesi, two notes per call in C. elisae). The two species also have non overlapping dominant frequencies (Fig. 8 and Table 5). The call of C. marcoreyesi is more similar to the one of C. buckleyi sensu stricto, however, there are differences in call duration and number of pulses present in a note, C. marcoreyesi having much shorter calls and fewer pulses/note (Fig. 8 and Table 5).

Distribution. Centrolene marcoreyesi is endemic to the eastern slopes of the southern Ecuadorian Andes (Fig. 7), where it is known from four localities within the Zamora Chinchipe Province: Estación Científica San Francisco, Abra de Zamora, Parque Nacional Podocarpus and Guarumales, at an altitudinal range of 1,840–2,190 m.

Conservation Status. We followed IUCN criteria to assess the current extinction risk of this species (Gärdenfors et al., 2001). Even if some of the known localities of Centrolene marcoreyesi are inside protected areas, and as such benefit from conservation measures, this species is threatened by degradation of its habitats, especially due to cattle farming, introduction of invasive exotic species and illegal and legal mining. Thus, we propose C. marcoreyesi to be considered as Endangered (EN) B1 a, b (i, iii) with an estimated extent of occurrence <100 km2.

Phylogenetics. Centrolene marcoreyesi is inferred monophyletic, with significant BI (posterior probability = 0.99) and ML (bootstrap = 98) supports. The new species is sister to C. sabini (Fig. 1), although support for this relationship is moderate (posterior probability = 0.82, bootstrap = 94).

Etymology. The species epithet “marcoreyesi” is a noun in genitive case, with the Latin suffix “i” (ICZN 31.1.2). With this species, we tribute Marco M. Reyes-Puig (Fig. S2), a notable herpetologist from the herpetology division of the Museo Ecuatoriano de Ciencias Naturales (now Instituto Nacional de Biodiversidad, INABIO). Marco was the original collector of this new species on a field campaign to Zamora Chinchipe. We honor his work and memory as a brother (Juan Pablo Reyes-Puig), sister (Carolina Reyes-Puig), and friends.

Remarks. Centrolene marcoreyesi (ZSFQ 4417 [MRy 547], ZSFQ 4418 [Mry 548]) corresponds to the species cited as “Centrolene buckleyi [Ca1]” by Amador et al. (2018).

Biogeographic history of the Centrolene buckleyi complex

The Dispersal-Vicariance biogeographical model (DIVALIKE; Table S3) was the one selected as best fitting. The MRCA of our calibrated Centrolene species tree probably originates in the northern Andes of Ecuador ~7 Ma (late Miocene, 95% HPD: 5.9–7.9 Ma) (Fig. 11). Subsequently, two main clades diverged, a first clade with northern Andean species that includes C. buckleyi sensu stricto, and another clade with northern and central Andean species that consists of the two new species described here C. elisae and C. marcoreyesi. The species of the second clade diverged shortly after at 5.7 Ma (late Miocene to early Pliocene, 95% HPD: 4.3–7.2 Ma), from ancestors that inhabited the northern Andes of Ecuador, Colombia, and Venezuela. The ancestral range of the clade formed by C. marcoreyesi and C. sabini diverged ~1.5 Ma (95% HPD: 0.2–3.1 Ma) and was probably during the Pleistocene in the north-central Andes. On the other hand, C. elisae and its sister species C. cf. elisae originated from an MRCA distributed only in the northern Andes and that diverged almost at the same time as C. marcoreyesi in the south, at >1.2 Ma (95% HPD: 0.4–2.2 Ma).

Figure 11 Ancestral ranges and rates of dispersal and vicariance under the DIVALIKE (Dispersal-vicariance) model inferred with the software BioGeoBEARS for Centrolene species.

Pie chart colors on the tips and nodes of the phylogeny correspond to the legend of the areas in the lower left. The colors, not represented in the map, correspond to a species or ancestor that is/was present in more than one region (e.g., Northern Andes Colombia/Ecuador, Northern Andes Venezuela/Colombia). The new species are depicted in bold. Prepared by Luis Amador.

Discussion

Centrolene buckleyi was recognized as a species complex based on acoustic and phylogenetic data (Guayasamin et al., 2006a, 2008, 2020; Amador et al., 2018). Here we redefine C. buckleyi sensu stricto and describe two new species based on molecular phylogenetics, morphological, acoustic, and osteological evidence. We highlight that the two new species are not sister to C. buckleyi sensu stricto, and that they exhibit a combination of traits that support their validity. As seen in other cryptic groups, the integration of different sources of traits is key for recognizing lineages, obscured by superficially similar external morphology and color patterns. In C. buckleyi sensu stricto, the shape of occipital condyles and its relation with exoccipital, shape, size and orientation of zygomatic and otic ramus in squamosals, and shape of anterior border of cultriforms process in parasphenoides, seem to be useful diagnostic features. We note, however, that our understanding of intraspecific variation in glassfrog osteological traits is limited and, therefore, the validity of these traits as diagnostic should be taken with caution.

Since vocalizations play a key role in intraspecific recognition (Wells & Schwartz, 2007), finding non-overlapping differences among closely related taxa reinforces the hypothesis that the lineages are, indeed, evolving independently (e.g., Centrolene buckleyi sensu stricto and C. elisae). Thus, even if somehow morphologically cryptic, species calls tend to exhibit more disparities, resulting in useful traits for species identification (Hutter & Guayasamin, 2012; Escalona et al., 2019; Köehler et al., 2017). Our findings of call differences in glassfrogs, are similar to recent studies have shown that call convergence, probably because of habitat filtering, is common (Mendoza-Henao et al., 2023).

Biogeography

The radiation of numerous amphibians is heavily influenced by the topographic complexity of the Andes that fosters allopatric speciation (Lynch & Duellman, 1997; Coloma et al., 2012; Páez-Moscoso & Guayasamin, 2012; Castroviejo-Fisher et al., 2014; Guayasamin et al., 2020). The genus Centrolene has been accumulating species in Neotropical mid-elevation habitats, long before the Andes reached their current elevations (Hutter, Guayasamin & Wiens, 2013). In this way, the peak diversification in Centrolene is recent, with most species originating during the late Miocene and Pliocene. The distribution and diversification patterns of the Centrolene buckleyi complex provide insights into the evolutionary history and diversification of glassfrogs. As shown in previous studies (Castroviejo-Fisher et al., 2014; Hutter, Guayasamin & Wiens, 2013, 2017; Guayasamin et al., 2020), speciation in Centrolene seems to be mediated mostly by the vicariant effect of the Andes, as well as by its linearity (Remsen, 1984; Graves, 1988; Guayasamin et al., 2020). The consequence of such a scenario is allopatric sister species, inhabiting very similar environments (i.e., niche conservatism) (Hutter, Guayasamin & Wiens, 2013) and, often, retaining morphological traits, as observed in the C. buckleyi complex.

The new species, Centrolene elisae, also fits the described linearity model, with divergent lineages occurring on the eastern Andean mountain chain (e.g., C. venezuelense from Venezuela, C. cf. venezuelense from Colombia, and C. elisae from Ecuador). Restriction of gene flow along the eastern slope of the Andes can have a topographic origin (e.g., Táchira Depression, Cofanes River basin, Quijos River basin), but could also be a consequence of random extinction and loss of contact zones (see Guayasamin et al.,2020).

The other new species, C. marcoreyesi and its sister species, C. sabini (both also distributed on the eastern slopes of the Andes) are separated by the Huancabamba Depression, which has also influenced the diversification of other Andean groups (Vuilleumier, 1969, 1984; Duellman, 1979; Fjeldså, Lambin & Mertens, 1999; Winger & Bates, 2015; Torres-Carvajal, Venegas & Sales Nunes, 2020; Venegas et al., 2021). This depression, a low-elevation, arid valley, likely represents a strong barrier for cloud forest species, adapted to constant humidity and cold climate (Hutter, Guayasamin & Wiens, 2013).

Pending taxonomic issues in Colombia

The Centrolene buckleyi complex still requires further work in Colombia. Our phylogenetic analysis included two populations from the Eastern Andean mountains near Bogotá, in the Department of Cundinamarca. These populations are closely related to C. venezuelense and C. elisae As detailed above, morphological and vocalization traits distinguish C. elisae from C. venezuelense. However, the identity of the Colombian populations remains uncertain. When comparing call attributes, we found differences in the frequency and duration of the calls recorded for the Colombian populations (KVV pers comm.) and those published for C. venezuelense (Señaris & Ayarzagüena, 2005), suggesting that both entities could represent different species. Yet, given the call differences are not as conspicuous as in other species (i.e., C. elisae), we emphasize the need to acquire additional data of Centrolene venezuelense. Taking this step will allow more robust comparisons of C. venezuelense with other entities within the C. buckleyi complex, including Colombian populations.

We also note that many more populations assigned to the Centrolene buckleyi complex in Colombia (Cochran & Goin, 1970; Ardila & Acosta, 2000; Lynch, 2001; Bernal, Páez & Vejarano, 2005; Amador et al., 2018; Guayasamin et al., 2020) could represent additional, undescribed species, considering that they are distributed on the three Andean cordilleras of Colombia. For example, although there is limited acoustic information on Colombian populations, calls from a population of C. cf. buckleyi in the Western Andean slope (Bolívar, Grant & Osorio, 1999) differ from those from the Eastern Andes studied here (KVV pers comm.), a result that may suggest that populations within the country correspond to independent taxonomic entities. Until further data are obtained, we suggest that Eastern Colombian populations should be treated as Centrolene cf. venezuelense.

Conservation

Centrolene elisae has been found in several private reserves (Chamanapamba Reserve, Yanayacu Biological Station, San Isidro Reserve), which have been key to preserving the forest on the Ecuadorian Amazonian slopes of the Andes. The Chamanapamba reserve, owned by EcoMinga Foundation is located in the Upper Pastaza river basin, that emerges as a relevant area for amphibian conservation, concentrating a high number of amphibian endemics (see Yánez-Muñoz, Cisneros-Heredia & Reyes-Puig, 2010; Reyes-Puig et al., 2010; Páez-Moscoso, Guayasamin & Yánez-Muñoz, 2011; Reyes-Puig & Yánez-Muñoz, 2012; Reyes-Puig, Reyes-Puig & Yánez-Muñoz, 2013; Reyes-Puig, Reyes-Puig & Ramírez-Jaramillo, 2014; Reyes-Puig et al., 2015, 2019, 2022a, 2022b).

For the other species described in this work, C. marcoreyesi, even considering that some populations are located inside Parque Nacional Podocarpus (one of Ecuador’s largest national parks) and the Key Biodiversity Area Abra de Zamora, the species faces several threats. The main identified threats are the loss and degradation of habitats due to cattle farming, the introduction of exotic species (Rainbow Trout, Oncorhynchus mykiss) and forest fires (Székely et al., 2020). To make matters worse, the spread of the illegal mining activities in Parque Nacional Podocarpus (Villa et al., 2022) and the increase of the mining concessions in southern Ecuador (Roy et al., 2018) threaten the survival of the species even in protected areas.

Abra de Zamora is a Key Biodiversity Area of unique importance due to the presence of many restricted range amphibian species and a center of amphibian diversification (Székely et al., 2020). Since 1938 until recently, 14 species of anurans were described from this relatively small area (e.g., Parker, 1938; Lynch, 1979; Trueb, 1979; Székely et al., 2020; Székely et al., 2023a) and others are waiting for the formal description (Paul Székely pers. comm.). From 2020, Abra de Zamora counts with a conservation action plan for amphibians (Ordóñez-Delgado et al., 2020), and several conservation projects were implemented by the EcoSs Lab group from the Universidad Técnica Particular de Loja in collaboration with Naturaleza y Cultura Internacional NGO, with the main purpose of safeguarding the unique ecosystems found here.

The other population of C. marcoreyesi, from Guarumales is located in the Sangay-Podocarpus connectivity corridor. Although this corridor is not formally part of the Ecuadorian system of protected areas, it incorporates a participatory model of management for conservation with the direct involvement of local governments (Sánchez-Nivicela, 2022). The area is threatened mainly by habitat destruction associated to the expansion of the agricultural/cattle raising frontier.

Conclusion

We provide an integrative analysis of the Centrolene buckleyi species complex and describe two new species, Centrolene elisae and C. marcoreyesi. Our phylogenetic hypothesis suggests that C. elisae is the sister species of C. venezuelense, and that C. marcoreyesi is the sister species of C. sabini. Speciation is most likely driven by the linearity of the Andes and the barriers formed by river valleys. We suggest that the two new species should be listed as Endangered.

Supplemental Information

Supplemental Information 1 Residuals obtained of morphologial measurementes to performed principal component analysis (PCA) and discriminant analysis of principal components (DAPC).

Supplemental Information 2 Estimates of genetic p-distances for the 16S mtDNA gene, calculated with the MEGA software.

Green: 0.0–3.0%, Blue: >3.0–5.0%, and Yellow >5.0%.

Supplemental Information 3 Comparison of biogeographic models tested with BioGeoBEARS for Centrolene species.

Model abbreviations are as follows: BAYAREALIKE = Bayesian inference of historical biogeography for many discrete areas (with likelihood interpretation); DEC = Dispersal-Extinction-Cladogenesis; DIVALIKE = Dispersal-Vicariance Analysis (with likelihood interpretation). LnL = Log-likelihood score; d = dispersal; e = extinction; AICc = standard correction to Akaike’s Information Criterion.

Supplemental Information 4 Species and genetic markers used in this study.

Sequences downloaded from Genbank are identified by their codes and sequences generated in this study are in bold.

Supplemental Information 5 Genetic p-distances among Centrolene species (16S).

Supplemental Information 6 The new species described herein are named after Elisa Bonaccorso (left) and Marco Reyes-Puig (right).

We are grateful to the people who provided specimens and tissues under their care for this study: Luis Coloma, Andrea Terán (Centro Jambatu), Mario H. Yánez-Muñoz, Miguel Urgilés (Instituto Nacional de Biodiversidad), Gabriela Maldonado, Mónica Páez (Museo de Zoología Universidad, Tecnológica Indoamérica), Mónica Guerra, Diego Almeida (Colección de Herpetología, Museo de Historia Natural “Gustavo Orcés V.” (MEPN-H), Escuela Politécnica Nacional), Juan C. Sánchez-Nivicela, Sebastián Valverde and to the Laboratory of Terrestrial Zoology and the Museum of Zoology, IBIOTROP Institute, Universidad San Francisco de Quito USFQ (Ecuador) for access to the specimens under their care, high magnification stereomicroscopy equipment and facilities, and the support provided by Emilia Peñaherrera, Mateo Davila, Carolina Reyes-Puig and David Brito. Cristina Paradela from the CT-Scan service of the Museo Nacional de Ciencias Naturales (MNCN), obtained the images for the osteological studies. Special thanks to Fernando Rojas-Runjaic for his accurate review comments and for the English review throughout the manuscript, to Marco Rada and Carolina Reyes for reading an early version of the manuscript and providing some suggestions. Technical support in the laboratory was provided by Gabriela Gavilanes (LBE), Mailyn Gonzalez and Eduardo Tovar Luque (Instituto Humboldt). DFM is grateful to Fundación Carolina and Universidad Rey Juan Carlos (URJC). Special thanks to Tito Recalde and Javier Robayo for their help during logistics and fieldwork (Chamanapamba reserve, EcoMinga Foundation), and José Simbaña (Yanayacu Biological Station), Jaime Culebras is grateful to Francesca Angiolani-Larrea, Steve Greenwood, Estelle Cheuk, Stuart Dunn, Gary Stadden, Daniel Hualpa, Santiago Hualpa, Sebastián Kohn and Darwin Núñez for their fieldwork help. Finally, Juan Pablo Reyes-Puig thanks to Miguel Urgilés Merchán, Gabriela Lagla, Cristian Páucar, Diego Inclán, Francisco Prieto, Ministerio del Ambiente, Parque Nacional Sangay, Angel Palacios, and Chistian Clavijo.

Additional Information and Declarations

Competing Interests

Author Contributions

Field Study Permissions

DNA Deposition

Data Availability

New Species Registration

The authors declare that they have no competing interests.

Daniela Franco-Mena conceived and designed the experiments, performed the experiments, analyzed the data, prepared figures and/or tables, authored or reviewed drafts of the article, and approved the final draft.

Ignacio De la Riva conceived and designed the experiments, performed the experiments, analyzed the data, authored or reviewed drafts of the article, and approved the final draft.

Mateo A. Vega-Yánez performed the experiments, analyzed the data, prepared figures and/or tables, authored or reviewed drafts of the article, and approved the final draft.

Paul Székely performed the experiments, analyzed the data, prepared figures and/or tables, authored or reviewed drafts of the article, and approved the final draft.

Luis Amador performed the experiments, analyzed the data, prepared figures and/or tables, authored or reviewed drafts of the article, and approved the final draft.

Diego Batallas performed the experiments, analyzed the data, authored or reviewed drafts of the article, and approved the final draft.

Juan P. Reyes-Puig analyzed the data, authored or reviewed drafts of the article, and approved the final draft.

Diego F. Cisneros-Heredia analyzed the data, authored or reviewed drafts of the article, and approved the final draft.

Khristian Venegas-Valencia analyzed the data, authored or reviewed drafts of the article, and approved the final draft.

Sandra P. Galeano analyzed the data, authored or reviewed drafts of the article, and approved the final draft.

Jaime Culebras performed the experiments, analyzed the data, authored or reviewed drafts of the article, and approved the final draft.

Juan M. Guayasamin conceived and designed the experiments, performed the experiments, analyzed the data, prepared figures and/or tables, authored or reviewed drafts of the article, and approved the final draft.

The following information was supplied relating to field study approvals (i.e., approving body and any reference numbers):

Our study was conducted under permits MAE-DNB-CM-2018-0105, MAE-DNB-CM-2015-0016, and MAATE-cmarg-2022-0575, issued by the Ministerio del Ambiente, Agua y Transición Ecológica (MAATE), Ecuador.

The following information was supplied regarding the deposition of DNA sequences:

The sequences are available at GenBank: OR479083.1–OR479128.1.

The following information was supplied regarding data availability:

The raw measurements and other raw data are available in the Supplemental Files.

The following information was supplied regarding the registration of a newly described species:

Publication LSID: urn:lsid:zoobank.org:pub:EB178068-646B-4071-96B4-C0614D90A366.

Centrolene elisae LSID: urn:lsid:zoobank.org:act:E9E154BD-3D98-471B-8A41-7F8870CA1572.

Centrolene marcoreyesi LSID: urn:lsid:zoobank.org:act:FC59A40D-6F9C-45BF-A6FE-280FA20BC2D1.

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
