# Peer review of "Simplifying the Centrolene buckleyi complex (Amphibia: Anura: Centrolenidae): a taxonomic review and description of two new species"

_PeerJ, doi:10.7717/peerj.17712_

## Round 0.1 · original submission · Major Revisions

Dear authors,

I'm sorry it look you so long to have a first decision from us. I have just been assigned this manuscript.

There are four reviews back now. While all of them saw your work with enthusiasm, they also pointed out many aspects that deserve the authors' attention. I also believe this could be a good contribution to centrolenid taxonomy, but I'd like to call the authors' attention to a few required data that are lacking in the results. Notice R3 was more critical, but provided a road map on how to improve the paper nevertheless. Pay attention to his/her comments on the lack of phylogenetic evidence for species status of one proposed species.

I have a couple more questions:

1) Delete L. 66-73 or move it to the end of the Introduction. In L. 74 delete "Glassfrogs and the Centrolene buckleyi species complex:"

2) L. 142-7: I second R1 comments on the PCA for morphometrics analysis. Consider using log-shape ratio (Mosimann 1970) to remove the effect of size on linear morphometrics traits. Actually, the whole analysis of morphometric variables has to be redone. It doesn't make sense to test for normality of data, much less as an evidence for diference between species. Remember that the assumption of linear models (Gauss-Markov theorem) are about the residuals, not the actual data. So, delete Table 3 and 4. But most importantly, it doesn't make sense to compute a PCA and then use separate, univariate t-tests for each variable. If your data are multivariate, you need to use multivariate stats. If you want to test for differences between putative species in several linear morphometric variables at once you need to use DAPC or some kind of related analysis. This will also provide you with the most important variables that distinguish species. Table 5, remember to translate machos/hembras.

3) L. 148-51: move this to figure legends.

4) L. 208-10 move to tables and figure legends.

5) why have you tested for differences between species in terms of morphometry, but not bioacoustics? The same analysis could be done, with DAPC.

6) What are the / below the species in Table 7, as in n=1/4/6/29?

7) In Table 8, provide delta AICc and Akaike weights.

8) in Fig. 4 use convex hulls instead of ellipses.

9) I agree with R4 in which the manuscript is extremely lengthy and looks more like a monograph. Authors have to make an effort to be as consice as possible, where it is possible. For example, delete L. 36-8, delete "discovered within this complex" in L. 42. Delete 42-44 (the diagnostic traits are listed down below already). You have a gargantuan number of figures. I recommend a re-organisation. I highly recommend you to shorten the osteological description to only present the significant differences between the three species, since most of the characteristics are common to the family. Move Fig. 5 to suppl mat. Combine Fig. 6 and 7 into a single plate. Delete Fig. 10 or combine it to Fig. 3. Combine Fig. 11 and Fig. 2. What's the point of having Fig. 3 and 12? Delete Fig. 12. Combine Fig. 13 and 14. Delete Fig. 17. Combine Fig. 18 and 3. Combine Fig. 19 and 2. Combine Fig. 20 and 21.

10) Cite software correctly, including MAFFT, Raven, R and R packages (type citation() and citation("packageName").

11) what's the difference between 'Definition' and 'Diagnosis' in L. 413 and 423 and L. 632 and 642? Can you combine those?

12) L. 572-80: "the dominant frequency is higher in C. elisae', higher than what? Also for this entire paragraph, which test have you conducted to say that a given call parameter is higher or lower?

13) L. 591-4 and 602-3 and 796-7 and 805-6. Delete. These are redundant.

14) L. 623-31: delete and change 403-412 to apply to both new species. Perhaps placing this paragraph before the two descriptions.

15) L. 779 Differs from whom?

Remove all figure and table citations from the Discussion.

Deposit CT-scan images in MorphoSource.

·

Basic reporting

Clear and unambiguous, professional English used throughout. Literature references, sufficient field background provided. Professional article structure, figures, tables. Raw data shared.

Experimental design

Original primary research within Aims and Scope of the journal. Research question well defined. The investigation must have been conducted rigorously and to a high technical standard. Methods described with sufficient detail & information to replicate.

Validity of the findings

All underlying data have been provided; they are robust, statistically sound, & controlled. Conclusions are well stated, linked to original research question & limited to supporting results.

Additional comments

Authors assessed and redefined the species boundaries of C. buckleyi under an integrative approach, and formally described two new species discovered within the C. buckleyi complex. Morphological traits, vocalizations, osteology, and genetic distances, all supported the validity of the new species. However, there are some questions need to be verified. (1) PCA analysis is unclear. Generally, to reduce the effect of allometry, you should calculate the ratio of each character to SVL and then log-transformed to better approximate the normal distribution. (2) Some figures contain subtitle (such as A, B, C) , but they are not cited in the text, please check. (3) Some others errors have been highlighted in the text.

Reviewer 2 ·

Basic reporting

This is an excellent description of two new species of Centrolene. The writing is generally clear and concise, though the article would benefit from some light editing to improve sentence structure and wording in a few places. There are many figures, but all are very well done and easy to interpret. The article is self-contained and appears to represent a coherent 'unit of publication.'

Experimental design

This study fills a knowledge gap by correcting inaccurate taxonomy that underestimates species-level diversity. Phylogenetic, acoustic, and morphometric analyses all appear to have been conducted rigorously and to a high technical standard, and are adequately described. I'm less familiar with the biogeographic analyses, but they are adequately described and seem to have been performed appropriately. The conclusions are well-stated, and importantly, limited to those supported by results (e.g., other authors may have described additional species based on the genetic data alone).

However, the Materials and Methods lacks this required paragraph (see https://peerj.com/about/policies-and-procedures/#new-species):

"The electronic version of this article in Portable Document Format (PDF) will represent a published work according to the International Commission on Zoological Nomenclature (ICZN), and hence the new names contained in the electronic version are effectively published under that Code from the electronic edition alone. This published work and the nomenclatural acts it contains have been registered in ZooBank, the online registration system for the ICZN. The ZooBank LSIDs (Life Science Identifiers) can be resolved and the associated information viewed through any standard web browser by appending the LSID to the prefix http://zoobank.org/. The LSID for this publication is: [INSERT HERE]. The online version of this work is archived and available from the following digital repositories: PeerJ, PubMed Central SCIE and CLOCKSS."

Validity of the findings

The authors convincingly show that the two species are in fact new, and that additional undescribed species exist.

Additional comments

comments

line 46: sounds like you're beginning a whole new paragraph or section; can you reorganize to make the paragraph flow more smoothly?
77: the Andes
105-6: is this statement necessary? I can't tell if the authors are trying to be funny, but I'd remove these two sentences.
143: missing 'analysis'
146: performed in R (not 'R Core Team')
600: how about education or training instead of 'formation', which sounds strange
894-5: need em dash, not hyphen
Table 1: left justify text
Figure 1: I think an isosceles triangle would look better (for the collapsed Nymphargus clade)
Figure 4: use better (more color-blind friendly) colors
Figure 5: take out numbers on species labels, abbreviate Centrolene in the labels, e.g. C. marcoreyesi sp. nov. and put the names, not numbers on the x-axis as well
Figure 6: Beautiful figure. But I think a sans serif font would be more clear (e.g., Arial or Helvetica)
Figure 17: is a figure with photographs of the scientists necessary? I'm not sure it adds anything to the article, except to make it longer.
Supplemental Table (no number): don't italicize sp. nov. and add periods
Table S1: missing a space in #16
Appendix 1: there are two spaces between the genus and species for some species; also, it would be useful to have the latitude and longitude fields in separate columns for easy parsing of the data and mapping by future workers. It would also be helpful to georeference samples without coordinates (e.g., using GEOlocate), but not necessary.

Reviewer 3 ·

Basic reporting

In general, the article is a good contribution to the knowledge of the diversity of glass frogs. However, the authors should pay a little more attention to their analyses, results, and obviously discussion and conclusions. There are several extremely important aspects that seem to have been poorly addressed: the justification for the selection of the inference method, the data that in some parts are treated as evidence and in other places as data only (e.g., osteology and calls). I am also concerned that the abstract mentions phylogenetic evidence for the new species, but at least for one of them the phylogenetic evidence does not corroborate the species.

Another striking part is that the authors argue that they have evidence to talk about geographic barriers to promote speciation in the Andes. It seems a bit presumptuous for the authors to think that they have evidence to discuss the types of speciation. The argument that the authors use goes from the particular (case of two new species) to a general (speciation in the Andes). If you have evidence to go from the general to the particular, you have evidence to talk about it. Otherwise, it is best to go as far as the evidence takes you, even if we are tempted to go further.

Based on these arguments, I suggest that the authors pay a little more attention to what they are proposing in their contribution, from the selection of analysis, data analysis, and discussion and conclusions.

Comments:

Line 43: If you obtain a polytomy, you cannot say that you have phylogenetic evidence to corroborate your new species (e.g., species 1).

Lines 44-45: Authors should stop using catch-all phrases to make articles eye-catching. In this article, there is no analysis in any way to prove what this line says.

Lines 68-75: Although it is valid to focus on the importance of knowing to conserve, I believe that it is also important to know to improve our understanding of diversity. In other words, the text as it is written unnecessarily limits the process of describing a species.

Line 166-167: The relevance of this is unclear. Would you like to see why?

LInes 169-170: It is becoming increasingly common to say what is done but not explain why it is done. Both methods are totally different in their a priori assumptions and therefore in the way in which evidence is analyzed.

In general, there are two reasons for selecting an inference method. The first is to simply follow the trend (although few people admit it). The other reason is to use theoretical/practical arguments. In scientific articles, it is important to explain why something is done or not done.

Lines 234-247: In line with the previous comment, why do you select ML over BI? It is implicit (not explicit) in your text that you select it because it has greater support in the clades. If that is the selection argument, mention it and in that way it will be clear to people the level of objectivity of your analysis.

Lines 596-599: If you obtain a polytomy, you cannot say that you have phylogenetic evidence to corroborate your new species.

Lines 835-836: For example?.
Also, the authors, studied only one specimen. What about the variation?

Lines 895-896: You should restructure your summary because you say that you have phylogenetic evidence, but what you have is a lack of evidence.

Line 897: What do you mean by internal? Are you referring to anatomy? Morphology and anatomy are different.

Experimental design

As mentioned in my review, the argument for performing several of the analyses is not clear. The authors should explain whether it is for operationalism or if there are arguments behind the selection.

Validity of the findings

In my opinion, some of the findings that the authors present are beyond the scope of their evidence.

Additional comments

The authors should pay more attention to what they are proposing. We must go as far as the evidence takes us and always avoid filler phrases and instead talk about the results we have.

Reviewer 4 ·

Basic reporting

This manuscript describes two new Centrolene species and redefines one taxon combining mtDNA, external morphology, osteology and bioacoustics as well as historical biogeography. This work is very thorough and overall well-written although a lot of minor things deserves clarification/improvements notably the grammar and the structure of the discussion. The lack of consistency and the presence of many typos indicates that the authors did not carefully revise the ms before submission. I probably missed a lot of them, so please really do it before the next submission. The methodology of the 3D osteology should be better described, as well as the biogeographic analysis, and the diagnoses which should mention only diagnostic traits (not overlapping morphometric values).
Because of its thoroughness, this ms is very long and I have to say that I couldn't find the time to look at the morphological descriptions and osteological parts with scrutiny.
One thing that is dearly missing is a rationale about how the authors identified recently collected typical C. buckleyi. I found no justification in the ms about type locality, type examination and similarity with fresh material.
There are too many figures I think, Some could be moved to supl mat and some could be fused.
Otherwise I would like to congratulate the authors for this important contribution that I think will reach the requirement for acceptance with few efforts.

Experimental design

OK

Validity of the findings

OK

Annotated reviews are not available for download in order to protect the identity of reviewers who chose to remain anonymous.

---

## Round 0.2 · Minor Revisions

I have now heard back from two of the reviewers from previous round. Like them I want to thank authors for engaging with both reviewer's and mine comments. However, there's still room for improvement in a few aspects. Please, consider R3 comments and the minor ones from R4.

After this last round of revision, I believe the paper will be in good shape for acceptance.

Reviewer 3 ·

Basic reporting

After reviewing the responses to the highlighted points, I believe some could benefit from further explanation. In some instances, the responses seemed to introduce unrelated points. For strong arguments, it's important to keep evidence lines independent; conclusions in one area shouldn't rely on evidence from another. I would appreciate seeing this approach applied more consistently in the responses. Additionally, some responses appear to rely on fallacies like bandwagon appeals (appealing to popularity) or ambiguity to support their claims. In conclusion, I believe several of the responses to the highlighted points could benefit from further explanation or elaboration. I have attached my comments on the reviewers' responses in this round of review.

Experimental design

The previous review identified several areas for improvement in this section. The authors' responses haven't fully addressed these issues yet.

Validity of the findings

The previous review identified several areas for improvement in this section. The authors' responses haven't fully addressed these issues yet.

Additional comments

For strong and publishable contributions, directly addressing reviewers' points and focusing arguments on the core discussion are key.

Annotated reviews are not available for download in order to protect the identity of reviewers who chose to remain anonymous.

Reviewer 4 ·

Basic reporting

This revised version requires only very minor edits (see annotated PDF)

Experimental design

ok

Validity of the findings

ok

Additional comments

NA

Annotated reviews are not available for download in order to protect the identity of reviewers who chose to remain anonymous.

---

## Round 0.3 · accepted · Accept

Thank you for carefully responding to the last round of reviewer's comments and modifying the text accordingly. I believe the paper is now ready to be published.